# How (Not) to Hybridize Neural and Mechanistic Models for Epidemiological Forecasting

**Yiqi Su** [1]   **Ray Lee** [1]   **Jiaming Cui** [1]   **Naren Ramakrishnan** [1]

## Abstract

Epidemiological forecasting from surveillance data is a hard problem and hybridizing mechanistic compartmental models with neural models is a natural direction. The mechanistic structure helps keep trajectories epidemiologically plausible, while neural components can capture non-stationary, data-adaptive effects. In practice, however, many seemingly straightforward couplings fail under partial observability and continually shifting transmission dynamics driven by behavior, waning immunity, seasonality, and interventions. We catalog these failure modes and show that robust performance requires making non-stationarity explicit: we extract multi-scale structure from the observed infection series and use it as an interpretable control signal for a controlled neural ODE coupled to an epidemiological model. Concretely, we decompose infections into trend, seasonal, and residual components and use these signals to drive continuous-time latent dynamics while jointly forecasting and inferring time-varying transmission, recovery, and immunity-loss rates. Across early outbreak and multi-wave regimes, our approach attains the lowest RMSE on all five datasets (up to 57% reduction over the strongest baseline), predicts the peak within one time step on four of five, and recovers time-varying epidemiological rates within ground-truth ranges, without relying on auxiliary covariates.

## 1. Introduction

Given historical epidemic curves, e.g., daily infections, hospitalizations, or deaths, the goal of epidemiological forecasting is to predict future trajectories. Epidemiologists have developed a broad range of models that have proven effective across outbreaks such as H1N1, Ebola, and most recently, COVID-19 (Cori & Kucharski, 2024).

Most forecasting models fall into three main families: compartmental models, statistical or machine learning models, and hybrid models. Compartmental models built on ordinary differential equations (ODEs), such as SIR (Susceptible-Infectious-Recovered) model, have long formed the foundation of epidemiological modeling (Anderson & May, 1992; Hethcote, 2000). However, they rely on strong assumptions about parameter stationarity (Lloyd, 2001). Meanwhile, statistical approaches such as ARIMA, state-space models, and Gaussian processes have also been widely used, particularly for short forecasting horizons (Box et al., 2015; Pivk & Le Diberder, 2005; Cori et al., 2023; Rasmussen, 2004). More recently, deep learning models, including recurrent neural networks, temporal convolutional networks, and transformer-based architectures have also achieved strong empirical performance (Hochreiter & Schmidhuber, 1997; Bai et al., 2018; Li et al., 2019; Zhou et al., 2021). Yet these models typically operate as black boxes, lack epidemiological interpretability, and may produce physically inconsistent forecasts for long horizons or under regime shifts (Shaman & Karspeck, 2012; Bracher et al., 2021). To address this issue, hybrid frameworks that embed neural components within compartmental models have been proposed, including neural ODEs (Chen et al., 2018), latent ODEs (Rubanova et al., 2019), and physics-informed neural networks (PINNs). In principle, hybrid models can combine mechanistic interpretability with data-driven flexibility; in practice, however, they remain fundamentally challenged by partial observability and non-stationarity.

**Partial observability.** Epidemiological data is inherently incomplete: typically only cases/deaths/hospitalizations (or proxies such as symptomatic rates) are observed, while other states (e.g., susceptible or exposed) are latent (Fairchild et al., 2018; Ryu et al., 2022). Such undetermined systems make both inference and forecasting ill-posed, since multiple latent trajectories can explain the same observations, and full-state supervision is rarely available.

**Non-stationarity.** Key epidemiological parameters (e.g., transmission or recovery rates) evolve over time due to climate, mobility, behavior change, policy interventions, and

---

[1]Department of Computer Science, Virginia Tech, Alexandria, USA. Correspondence to: Naren Ramakrishnan <naren@vt.edu>.

*Proceedings of the 43rd International Conference on Machine Learning*, Seoul, South Korea. PMLR 306, 2026. Copyright 2026 by the author(s).

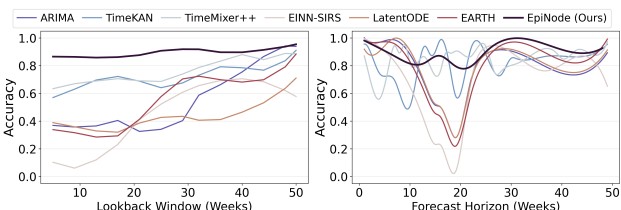

*Figure 1.* EpiNode accuracy as a function of lookback window and forecast horizon.

accumulated immunity (Keller et al., 2022). For example, influenza transmission varies across seasons and between waves, and retrospective analyses show that assuming constant parameters can induce systematic bias and degrade peak predictions (Castro Blanco et al., 2024). Modeling time-varying parameters is therefore essential, but increases estimation complexity from $O(n)$ to $O(nT)$.

Crucially, these challenges interact and amplify each other: latent states complicate parameter estimation, while parameter misestimation feeds back into incorrect latent-state reconstruction. This vicious cycle worsens with longer forecast horizons. While short-term forecasts benefit from temporal autocorrelation, accuracy deteriorates rapidly as the horizon grows: errors compound due to uncertainty in latent compartments and evolving dynamics. The effect is especially pronounced near epidemic peaks, where small growth-rate errors translate into large peak-timing and peak-magnitude errors (Antulov-Fantulin & Böttcher, 2022). This motivates methods that improve long-horizon stability and peak predictability and a careful accounting of when 'hybridization' helps versus hurts.

Our key contributions are:

- We characterize and demonstrate key failure modes in neural–mechanistic hybrid epidemic models, highlighting how the wrong architecture can destabilize long-horizon rollouts and bias peak predictions.

- We propose EpiNode, a decomposition-controlled hybrid framework that couples a neural ODE with mechanistic SIRS dynamics to address partial observability and non-stationarity by explicitly modeling the multiscale structure of the observed infection signal. The mechanistic SIRS layer enforces epidemiological consistency and supports stable long-horizon rollouts (Figure 1). The neural component recovers time-varying transmission, recovery, and immunity-loss rates within bounded, interpretable ranges.

- We demonstrate the broad applicability of the proposed model to seasonal epidemics across outbreak onset, peak, and post-peak phases, through extensive synthetic and real-world benchmarks.

## 2. Failure Modes

We begin by outlining failure modes specific to epidemiological modeling when using neural ODEs and related machine learning approaches. These modes can be viewed as 'antipatterns', i.e., what not to do.

### 2.1. Neural ODEs fail under partial observability even when they fit observed infections.

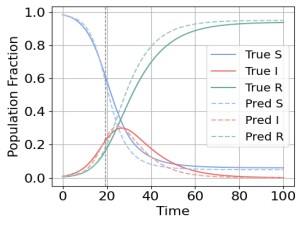 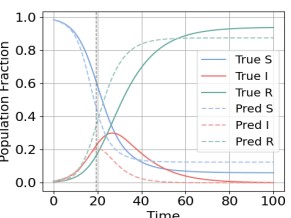

*(a)* Neural ODE with full SIR  *(b)* Neural ODE with I-only

*Figure 2.* Failure of SIR forecasts from neural ODEs under full and partial observability at a train/forecast split of $0.2/0.8$.

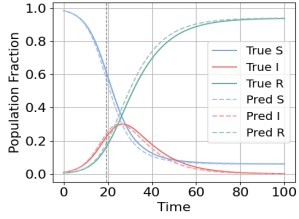 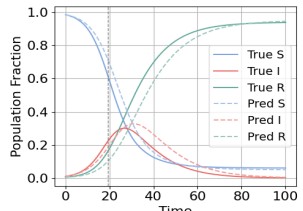

*(a)* AE-NODE with full SIR  *(b)* AE-NODE with I-only

*Figure 3.* Failure of SIR forecasts from AE-NODE under full and partial observability at a train/forecast split of $0.2/0.8$ with full-state supervision (a) and under I-only supervision (b).

We first evaluated vanilla neural ODEs (Chen et al., 2018) and autoencoder-based neural ODE (AE-NODE) pipelines, wherein an encoder mapping observations to a latent state, followed by continuous-time latent evolution, and a decoder. These models perform well when all compartments are observed (Figure 2a, Figure 3a), but their performance degrades substantially when only the infected trajectory I(t) is available (Figure 2b, Figure 3b). (This is the most reasonable assumption since infections can be assimilated from hospitalizations and other data, but other compartments are not observed.) Under partial observability, the encoder–decoder formulation introduces a non-identifiability issue: multiple latent trajectories and parameter configurations can explain the same observed I(t). As a result, the latent dynamics can drift in unobserved dimensions while still fitting short-term observations, leading to unstable long-horizon rollouts. This highlights that latent continuous-time modeling alone is insufficient without additional structure to anchor the latent state.

## 2.2. Bidirectional (adjoint-based) objectives fail to resolve identifiability and can still learn implausible latent dynamics.

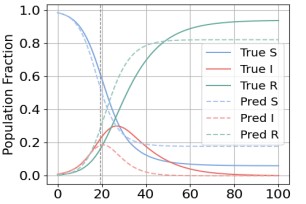 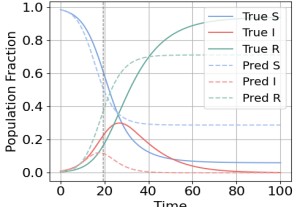

*(a)* AE-NODE with full SIR + bidirectional training

*(b)* AE-NODE with I-only supervision + bidirectional training

*Figure 4.* Failure of SIR forecasts from AE-NODE under short training windows, with bidirectional training.

Inspired by Koopman autoencoders (Azencot et al., 2020) and adjoint sensitivity analysis in neural ODEs (Chen et al., 2018), we explored bidirectional training objectives. The adjoint formulation makes it possible to efficiently compute gradients with respect to initial conditions and model parameters by integrating an auxiliary adjoint system backward in time, effectively enabling a backward-in-time learning signal that complements forward rollout (see architecture in the Appendix; Fig A1).

While adjoint-based training improves optimization stability and encourages temporally consistent latent dynamics, it does not address the fundamental identifiability challenge arising from partial observability (Figure 4a). When only the infected trajectory I(t) is observed (Fig. 4b), backward gradient propagation constrains the latent dynamics to be self-consistent but does not introduce new information about unobserved compartments or time-varying drivers.

Moreover, epidemic processes are intrinsically forward-evolving, governed by causal transmission and recovery mechanisms. However, in practice, bidirectional objectives allow the model to satisfy forward and backward consistency in latent space without learning epidemiologically meaningful dynamics. This suggests that optimization refinements alone cannot compensate for missing structural cues in the input signal. As a result, the model can satisfy both forward and backward objectives while still encoding implausible latent trajectories. This highlights that adjoint-enabled bidirectional learning is a powerful optimization tool, but must be complemented with structured, forward-driving signals to reliably learn interpretable and stable epidemic dynamics.

## 2.3. Physics-informed losses fail under sparse supervision and time-varying parameters by admitting degenerate solutions.

We next incorporated physics-informed losses (Sholokhov et al., 2023) that penalize violations of SIR/SIRS differential equations during training. These formulations aim to improve physical feasibility and short-term accuracy when

strong prior knowledge is available.

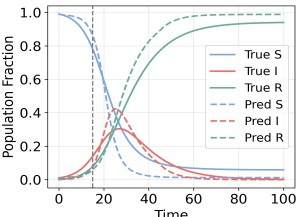 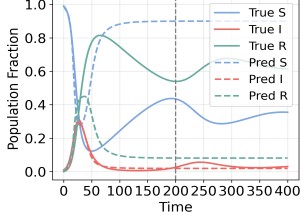

*(a)* Physics-informed loss with I-only under SIR

*(b)* Physics-informed loss with I-only under SIRS

*Figure 5.* Incorporating physics informed losses. (a) with only $I(t)$ observed, forecasts are reasonable with an SIR model assumption (a) but are poor under an SIRS model (b).

With only I(t) observed, under the basic SIR model (Figure 5a), forecasts remain qualitatively reasonable despite partial observability. Under the more expressive SIRS model (Figure 5b), the same supervision leads to severe errors in unobserved compartments and long-horizon dynamics, highlighting a fundamental identifiability gap when recovery and reinfection processes are not directly constrained.

## 2.4. Neural CDE and CDE–ODE hybrids fail to capture multi-wave dynamics.

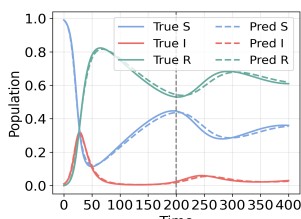 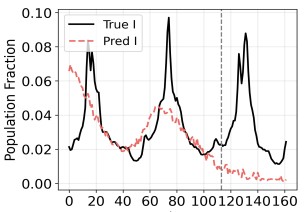

*(a)* NCDE-ODE with I-only under SIRS

*(b)* NCDE-ODE with I-only under ILI

*Figure 6.* Failure of Neural CDE-ODE forecasts on real data under partial observability. Forecasts are good for SIRS model (a), but fails to capture trends for ILI data (b).

Neural controlled differential equations (CDEs) (Kidger et al., 2020) were evaluated as a way to incorporate the observed signal as a continuous control path. We designed a NCDE-ODE architecture (Figure A2 in the Appendix), which captures synthetic SIRS generated using time-fixed parameters effectively (Figure 6a), but struggle with SIRS and multi-wave influenza-like illness (ILI) data (Figure 6b). This limitation arises because multi-wave dynamics are driven by latent forcing processes, such as seasonality and immunity waning that are not explicitly represented in the input channel. When only I(t) is provided, the model must implicitly infer these drivers, leading to poor extrapolation across epidemic waves.

## 3. EpiNode Framework

To address the above issues, we present **EpiNode** (Figure 7), a hybrid neural-physical framework for epidemic forecasting from partial observations. EpiNode integrates multi-scale signal decomposition, controlled latent continuous-time dynamics, and mechanistic SIRS evolution to jointly forecast epidemic trajectories and infer time-varying epidemiological parameters.

**Problem setup.** Let $\{(t_i, I_i)\}_{i=0}^{T-1}$ denote an epidemic time series, where only the infected compartment $I(t)$ is observed at discrete times $t_i$. The latent epidemic state is

$$\mathbf{y}(t) = [S(t), I(t), R(t)]^\top \in \mathbb{R}^3_{\geq 0}, \qquad (1)$$

with unknown, time-varying parameters $\beta(t)$ (transmission), $\gamma(t)$ (recovery), and $\delta(t)$ (immunity waning).

We assume $I(0)$ is known and initialize

$$S(0) = 1 - I(0), \qquad R(0) = 0. \qquad (2)$$

Our objective is to forecast $I(t)$ beyond the observation window, accurately predict epidemic peaks, and recover interpretable parameter trajectories.

**Variational mode decomposition (VMD).** Given a real-valued epidemic signal $x(t)$, we use VMD (Dragomiretskiy & Zosso, 2014) to decompose it into $K$ intrinsic mode functions $\{u_k(t)\}_{k=1}^K$, each associated with a center frequency $\omega_k$, by minimizing the total bandwidth of the modes subject to exact signal reconstruction. The constrained variational problem is defined as:

$$\min_{\{u_k\}, \{\omega_k\}} \quad \sum_{k=1}^K \left\| \partial_t \left[ \left( \delta(t) + \frac{j}{\pi t} \right) * u_k(t) \right] e^{-j\omega_k t} \right\|_2^2$$

$$\text{s.t.} \quad \sum_{k=1}^K u_k(t) = x(t), \qquad (3)$$

where $(\delta(t) + \frac{j}{\pi t}) * u_k(t)$ denotes the analytic signal of $u_k(t)$ obtained via the Hilbert transform, $\partial_t$ is the temporal derivative, and $*$ denotes convolution. This objective encourages each mode to be compact around its center frequency while collectively reconstructing the original signal.

**Augmented Lagrangian formulation.** The constrained problem in (3) is solved using the alternating direction method of multipliers (ADMM) by forming the augmented Lagrangian:

$$\mathcal{L}(\{u_k\}, \{\omega_k\}, \lambda)$$

$$= \alpha \sum_{k=1}^K \left\| \partial_t \left( (\delta(t) + \tfrac{j}{\pi t}) * u_k(t) \right) e^{-j\omega_k t} \right\|_2^2$$

$$+ \left\| x(t) - \sum_{k=1}^K u_k(t) \right\|_2^2 \qquad (4)$$

$$+ \left\langle \lambda(t), x(t) - \sum_{k=1}^K u_k(t) \right\rangle.$$

where $\lambda(t)$ is the Lagrange multiplier and $\alpha > 0$ controls the bandwidth penalty.

**ADMM updates.** Let $\hat{x}(\omega)$, $\hat{u}_k(\omega)$, and $\hat{\lambda}(\omega)$ denote the Fourier transforms of $x(t)$, $u_k(t)$, and $\lambda(t)$, respectively. The ADMM updates admit closed-form solutions in the frequency domain. At iteration $n + 1$, the mode update is

$$\hat{u}_k^{n+1}(\omega) = \frac{\hat{x}(\omega) - \sum_{i \neq k} \hat{u}_i^n(\omega) + \frac{1}{2}\hat{\lambda}^n(\omega)}{1 + 2\alpha (\omega - \omega_k^n)^2}, \qquad (5)$$

and the center frequency is updated as the energy-weighted mean frequency:

$$\omega_k^{n+1} = \frac{\int_0^\infty \omega \left| \hat{u}_k^{n+1}(\omega) \right|^2 d\omega}{\int_0^\infty \left| \hat{u}_k^{n+1}(\omega) \right|^2 d\omega}. \qquad (6)$$

The Lagrange multiplier is updated by

$$\hat{\lambda}^{n+1}(\omega) = \hat{\lambda}^n(\omega) + \tau \left( \hat{x}(\omega) - \sum_{k=1}^K \hat{u}_k^{n+1}(\omega) \right), \qquad (7)$$

where $\tau > 0$ is the dual ascent step size. Iterations continue until convergence. VMD produces an ensemble of band-limited modes with distinct frequency characteristics. In our framework, these modes are representing for three level of frequency levels: (i) trend $T(t)$ captures slow shifts driven by accumulated immunity, behavioral drift, policy interventions, and pathogen evolution, dominating long-horizon accuracy because trend errors compound over time; (ii) seasonal $S(t)$ captures periodic forcing from climate, school terms, and indoor contact patterns at annual or semi-annual scales, critical for peak-timing prediction; and (iii) residual $R(t)$ absorbs high-frequency reporting artifacts, super-spreader fluctuations, and stochastic dynamics, keeping the trend and seasonal estimates clean and stable. The decomposed modes are used as structured forcing signals for downstream epidemic modeling:

$$I(t) = T(t) + S(t) + R(t), \qquad (8)$$

where $T(t)$ captures low-frequency trends, $S(t)$ captures periodic or seasonal structure, and $R(t)$ represents residual

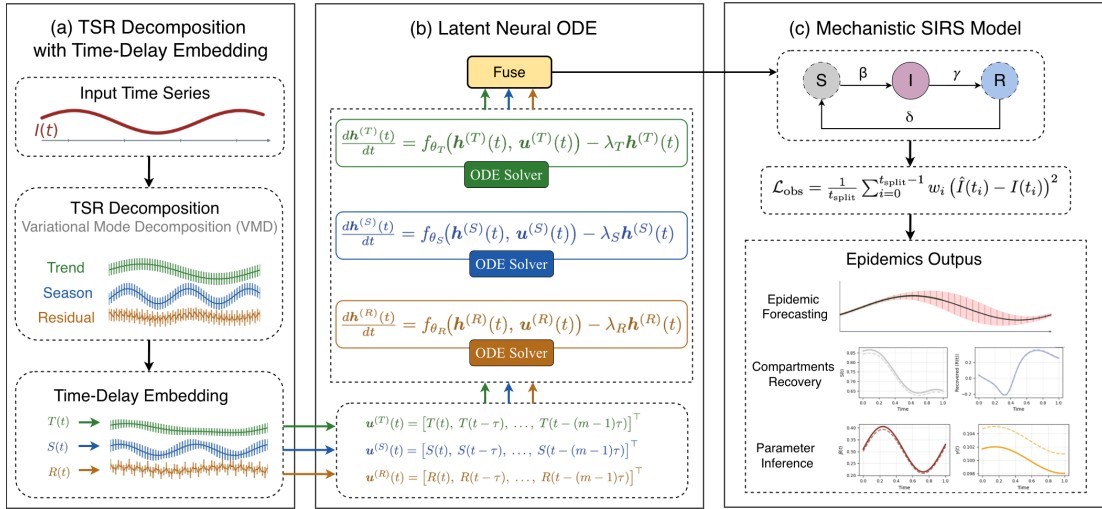

*Figure 7.* Overview of the proposed **EpiNode** framework. The observed epidemic time series is decomposed into *trend*, *seasonal*, and *residuals* components, which serve as multi-scale control signals for latent Neural ODEs. The fused latent representation is decoded into time-varying epidemiological parameters and coupled with mechanistic SIRS dynamics to produce physically consistent forecasts and interpretable parameter trajectories.

fluctuations. This decomposition isolates distinct temporal scales that would otherwise be confounded in a single observation stream.

**Channel-aware extrapolation past the observation window.** Equation (8) holds only where $I(t)$ is observed, i.e. on $[0, t_{\text{split}})$. To make the channels available to downstream components on the full window, we extend each channel into $[t_{\text{split}}, T)$ by a rule matched to its spectral character (Harvey, 1989; Cleveland et al., 1990). The trend is extrapolated as a linear drift fit by least squares on the last $L_{\text{fit}}$ training samples (Zeng et al., 2023):

$$T(t) \; = \; a\,t + b, \tag{9}$$

where $t \geq t_{\text{split}}$ and $(a, b)$ minimize $\sum_{j=t_{\text{split}}-L_{\text{fit}}}^{t_{\text{split}}-1} \big(T(t_j) - a\,t_j - b\big)^2$. The seasonal channel is tiled at its dominant period $p_S$, obtained from a periodogram of $\{S(t_j)\}_{j=0}^{t_{\text{split}}-1}$ (Bloomfield, 2000), yielding a seasonal-naive forecaster at the data-driven period (Hyndman & Athanasopoulos, 2021):

$$S(t) \; = \; S\big(t - p_S\,k(t)\big), \quad k(t) = \big\lceil (t - t_{\text{split}} + 1)/p_S \big\rceil, \tag{10}$$

where the RHS argument lies in $[t_{\text{split}} - p_S, \, t_{\text{split}})$ and is therefore defined by the causal VMD output. The residual is high-frequency signal with no extrapolable structure:

$$R(t) \; = \; 0. \tag{11}$$

**Time-delay embedding of control signals.** To provide temporal context and improve identifiability, we optionally apply a time-delay embedding independently to each component. For a component $x(t) \in \{T(t), S(t), R(t)\}$, corresponding to trend, seasonality, and residual respectively, we construct the lag-augmented control:

$$\mathbf{u}^{(x)}(t) = \big[x(t), x(t - \tau), \dots, x(t - (m - 1)\tau)\big]^{\top} \in \mathbb{R}^m, \tag{12}$$

where $\tau$ is the delay and $m$ is the embedding dimension. Each delay-embedded control $\mathbf{u}^{(x)}(t)$ is then provided to its own latent Neural ODE, allowing the model to capture component-specific temporal dependencies before fusion. When delay embedding is disabled, the control reduces to the instantaneous signal, i.e., $\mathbf{u}^{(x)}(t) = x(t)$.

**Collaborative latent neural ODEs.** We maintain three latent states corresponding to the TSR components:

$$\mathbf{h}^{(T)}(t) \in \mathbb{R}^{d_T}, \quad \mathbf{h}^{(S)}(t) \in \mathbb{R}^{d_S}, \quad \mathbf{h}^{(R)}(t) \in \mathbb{R}^{d_R}. \tag{13}$$

Each latent state evolves according to a Neural ODE:

$$\frac{d\mathbf{h}^{(c)}(t)}{dt} = f_{\theta_c}\big(\mathbf{h}^{(c)}(t), \mathbf{u}^{(c)}(t)\big) - \lambda_c\,\mathbf{h}^{(c)}(t), \tag{14}$$

where $f_{\theta_c}$ is a neural vector field, $c \in \{T, S, R\}$, $\mathbf{u}^{(c)}(t)$ is the TSR-based control input from (12), and $\lambda_c \geq 0$ is a learnable damping coefficient implemented via $\text{softplus}(\cdot)$. The ODEs are numerically integrated between observation times using an explicit solver (RK4 in our experiments).

**Latent fusion and parameter decoding.** The latent states are fused into a shared representation:

$$\mathbf{h}(t) = \text{Fuse}\Big(\big[\mathbf{h}^{(T)}(t); \mathbf{h}^{(S)}(t); \mathbf{h}^{(R)}(t)\big]\Big), \tag{15}$$

where $\mathrm{Fuse}(\cdot)$ is a multilayer perceptron. From $\mathbf{h}(t)$, we decode epidemiological parameters using a bounded parameter network:

$$(\tilde{\beta}(t), \tilde{\gamma}(t), \tilde{\delta}(t)) = g_\phi(\mathbf{h}(t)), \qquad \tilde{\cdot} \in (0, 1), \qquad (16)$$

followed by affine scaling into disease-specific ranges:

$$\begin{aligned}
\beta(t) &= \beta_{\min} + (\beta_{\max} - \beta_{\min})\tilde{\beta}(t), \\
\gamma(t) &= \gamma_{\min} + (\gamma_{\max} - \gamma_{\min})\tilde{\gamma}(t), \qquad (17) \\
\delta(t) &= \delta_{\min} + (\delta_{\max} - \delta_{\min})\tilde{\delta}(t).
\end{aligned}$$

**Evolving epidemic trajectories.** Given the decoded parameters, the epidemic state evolves according to the SIRS equations:

$$\begin{aligned}
\dot{S}(t) &= -\beta(t)\, S(t)\, I(t) + \delta(t)\, R(t), \\
\dot{I}(t) &= \beta(t)\, S(t)\, I(t) - \gamma(t)\, I(t), \qquad (18) \\
\dot{R}(t) &= \gamma(t)\, I(t) - \delta(t)\, R(t).
\end{aligned}$$

We integrate (18) using an RK4 step with $\Delta t_i = t_{i+1} - t_i$. After each step, we enforce non-negativity and approximate mass conservation:

$$\mathbf{y}(t_{i+1}) \leftarrow \alpha \frac{\mathbf{y}(t_{i+1})}{\|\mathbf{y}(t_{i+1})\|_1} + (1 - \alpha)\mathbf{y}(t_i), \qquad (19)$$

with $\alpha = 0.9$ in all experiments. The predicted infection trajectory is $\hat{I}(t) = (\mathbf{y}(t))_2$.

**Training objective.** Let $t_{\mathrm{split}}$ denote the end of the observation window. We train EpiNode end-to-end by minimizing a weighted mean squared error on the infected compartment:

$$\mathcal{L}_{\mathrm{obs}} = \frac{1}{t_{\mathrm{split}}} \sum_{i=0}^{t_{\mathrm{split}}-1} w_i \big(\hat{I}(t_i) - I(t_i)\big)^2, \qquad (20)$$

where $w_i$ increases linearly near the end of the training window to emphasize alignment at the forecast boundary. Gradients are backpropagated through all components, including the ODE solvers.

# 4. Experimental Results

Our experiments are intended to answer the below questions:

(1) **Performance evaluation (Section 4.5, Section H.1)**

    (a) **Forecasting accuracy.** How accurately does the model predict the full infection trajectory over time? (Section 4.5.1, Appendix H.1.1)

    (b) **Peak errors.** How accurately does the model predict the magnitude and timing of the infection peak? (Section 4.5.2, Appendix H.1.2)

(2) **Applications (Section 4.6, Appendix H.2)**

    (a) **Parameter inference.** Is the model able to recover meaningful and smooth trajectories of time-dependent epidemiological parameters? (Section 4.6.1 Appendix H.2.1)

    (b) **Regional dynamics.** Is EpiNode able to capture regional dynamics of infectious disease transmission? (Appendix H.2.2)

(3) **Ablation analysis (Appendix H.3)**

    (a) **Decomposition *vs* non-decomposition.** Does decomposing the observed infection time series improve modeling and forecasting performance compared to using the raw signal directly? (Appendix H.3.1)

    (b) **Effect of decomposition order.** How does the number of decomposed components affect performance? (Appendix H.3.2)

    (c) **Decomposition methods.** How do different decomposition choices affect model accuracy? (Appendix H.3.3)

    (d) **Contribution of time delay.** How does applying a time delay to the decomposed components influence prediction accuracy and learned dynamics? (Appendix H.3.4)

## 4.1. Ablation Procedures

**Single latent ODE.** We replace the three collaborated Neural ODEs in (14) with a single latent state $\mathbf{z}(t)$ as

$$\frac{d\mathbf{z}(t)}{dt} = f_\theta\big(\mathbf{z}(t), [\mathbf{u}^{(T)}(t); \mathbf{u}^{(S)}(t); \mathbf{u}^{(R)}(t)]\big) - \lambda\mathbf{z}(t), \qquad (21)$$

followed by the same fusion, parameter decoding, and SIRS rollout. This variant isolates the benefit of disentangling multi-scale dynamics into separate latent flows.

**Number of decomposed signals.** To study the impact of the number of decomposition components, we compare the accuracy over 1-component (1C) *vs* 2C *vs* 3C. In the case of 1C, the input time-series data will not be decomposed and thus this reverts to a vanilla neural ODE.

**Choice of decomposition methods.** We further examine which decomposition methods are most effective at producing band-limited components with well-separated frequency content. The decomposition techniques evaluated include MA (moving average, (Zhang et al., 2022) ), STL (Seasonal–Trend decomposition using Loess, (Ouyang et al., 2021)), VMD, Wavelet (Michau et al., 2022), SSA–VMD (Gao et al., 2023), and Neural Koopman–based approaches (Takeishi et al., 2017), covering a broad range of design philosophies (see details in Appendix C).

**Time delay.** When delay embedding is disabled, $\mathbf{u}^{(x)}(t) = x(t)$.

### 4.2. Benchmarking Models

We compare against a diverse set of baselines, including (1) classical statistical and nonlinear sequence baselines (ARIMA, RNN-based models (LSTM)), (2) state-of-the-art univariate forecasters that leverage decomposition or multiscale mixing to capture long-range temporal patterns (TimeKAN (Huang et al., 2025) and TimeMixer++ (Wang et al., 2025)), (3) a physics-informed neural model (EINN (Rodríguez et al., 2023)), (4) ODE-based continuous-time models (Neural ODE (Chen et al., 2018), Latent ODE (Rubanova et al., 2019), and KAN-ODE (Koenig et al., 2024)), and (5) a graph-based neural ODE model (EARTH (Wan et al., 2025)) (see details in Appdendix D). Specifically, as EINN employs SEIRm physics, we adapt the framework in two ways: a) replace SEIRm with SIRS, b) predict I rather than m. In both options, I is the only observed compartment. For EARTH, since we are focused on single region forecasting, we treat the single region as a graph with $N = 1$ node and set adjacency $A = [1]$. All models are evaluated under a *single-variate input* setting.

### 4.3. Datasets

#### 4.3.1. SYNTHETIC DATASETS

**SIRS with time-fixed and time-varying parameters.** We generate synthetic epidemics using the SIRS model under multiple parameter regimes: (1) in the fixed setting (SIRS (Fixed)), transmission ($\beta$), recovery ($\gamma$), and immunity-loss ($\delta$) rates remain constant over time, serving as a baseline for identifiability under stationarity; in the time-varying setting (SIRS (Varying)), parameters evolve periodically to emulate seasonal forcing, capturing recurring epidemic patterns.

**Mismatched epidemic physics.** To assess robustness to physics mismatch, we simulate data from alternative compartmental models: (1) the SIR setting removes immunity waning, testing the model's ability to adapt when the assumed SIRS structure is over-parameterized; (2) the SEIRS setting introduces an exposed compartment, increasing latent-state complexity and evaluating performance when the true dynamics deviate from the assumed model class.

#### 4.3.2. REAL-WORLD DATASETS

We use weekly Influenza-like illness (ILI) surveillance data collected by the U.S. Centers for Disease Control and Prevention (CDC) (available at `https://gis.cdc.gov/grasp/fluview/fluportaldashboard.html`) from all 10 U.S. Department of Health and Human Services (HHS) regions (Week 30, 2022–Week 30, 2025).

### 4.4. Evaluation Protocol

We evaluate all models under a unified protocol: **(1) Forecast accuracy:** evaluate forecasting performance using standard pointwise error metrics on the infection trajectory. **(2) Peak detection accuracy:** evaluate peak detection performance by comparing the predicted and true peak values (magnitude) and the predicted and true peak times (timing). For multi-wave sequences, we focus on the dominant peak within the forecasting window. **(3) Parameter estimation:** for models that infer epidemiological parameters, evaluate the quality of the estimated parameter trajectories.

**Evaluation metrics.** For forecast accuracy we report root-mean-square error (RMSE) on $I(t)$ over the entire forecast window and selected horizon windows. For peak detection we report signed and absolute errors for both peak timing and peak magnitude. For parameter inference on synthetic datasets, where ground truth is available, we report per-parameter RMSE for $\beta(t)$, $\gamma(t)$, and $\delta(t)$. All metrics are reported as mean $\pm$ std over random seeds to ensure statistical robustness.

### 4.5. Performance Evaluation

#### 4.5.1. FORECASTING ACCURACY

Figure 8 compares forecast trajectories across synthetic and real datasets at representative train/forecast splits. Table 1 reports the corresponding RMSE means and standard deviations and EpiNode attains the lowest mean RMSE in every column. On synthetic SIRS data with fixed parameters (Figure 8a), while most models achieve reasonable short-term accuracy, EpiNode maintains stable long-horizon rollouts without drift in unobserved compartments (RMSE 0.0023 *vs* 0.0046 for the next-best LatentODE). In the time-varying SIRS setting (Figure 8b), which introduces periodic forcing and immunity waning, the gap between EpiNode and baselines widens. Pure Neural ODE, Latent ODE, and KAN-ODE struggle to extrapolate across multiple waves, while physics-informed EINN exhibits biased peak magnitudes due to limited flexibility in handling non-stationary parameters. EpiNode reaches RMSE 0.0210 against 0.0247 for the strongest baseline EINN-SIRS. On physics-mismatched settings (SEIRS in Figure 8d and SIR in Figure A3b), EpiNode remains stable despite under- or over-parameterization of the assumed SIRS model, reaching RMSE 0.0195 on SIR (33% below EINN-SIRS) and 0.0107 on SEIRS, where LatentODE is the closest baseline at 0.0111.

For real-world datasets (seasonal ILI), Figures 8e, 8f show that EpiNODE consistently outperforms strong univariate baselines (TimeKAN, TimeMixer++) and continuous-time models in long-horizon accuracy. In particular, EpiNODE exhibits superior stability during post-split rollouts, avoiding the oscillatory or mean-reverting failures observed in purely

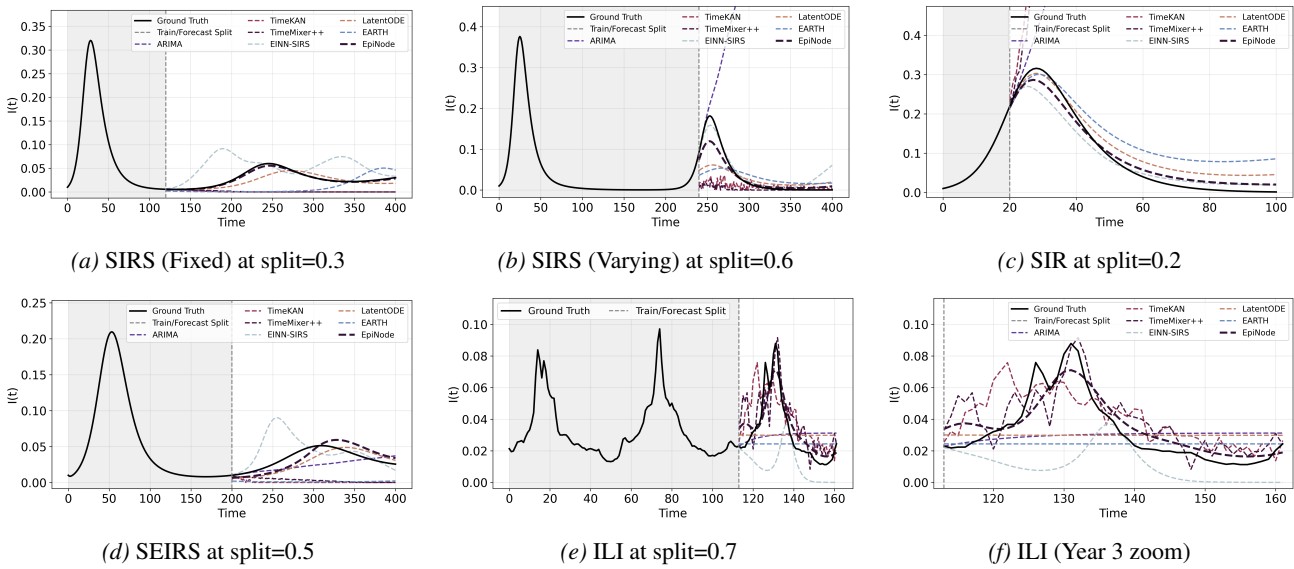

*Figure 8.* Forecast comparison on synthetic and real datasets at the headline train/forecast split per dataset.

*Table 1.* Benchmark comparison across synthetic and real datasets (mean $\pm$ std).

| Method | SIRS (Fixed) | SIRS (Varying) | SIR (Fixed) | SEIRS (Fixed) | ILI |
|---|---|---|---|---|---|
| ARIMA | 0.0322 | 0.8179 | 1.8492 | 0.0146 | 0.0216 |
| LSTM | 0.0324 (0.0000) | 0.1310 (0.0130) | 0.1921 (0.0043) | 0.0815 (0.0091) | 0.0224 (0.0045) |
| EINN-SIRS | 0.0345 (0.0031) | 0.0247 (0.0049) | 0.0291 (0.0014) | 0.0295 (0.0047) | 0.0302 (0.0033) |
| EINN-SEIRm | 0.0752 (0.0308) | 0.1038 (0.0290) | 0.1261 (0.0158) | 0.0563 (0.0236) | 0.0425 (0.0194) |
| NeuralODE | 0.0269 (0.0102) | 0.0497 (0.0103) | 0.0850 (0.0057) | 0.0307 (0.0098) | 0.0299 (0.0069) |
| LatentODE | 0.0046 (0.0039) | 0.0405 (0.0036) | 0.0353 (0.0062) | 0.0111 (0.0093) | 0.0228 (0.0035) |
| KAN-ODEs | 0.0637 (0.0002) | 0.0564 (0.0005) | 0.3918 (0.3648) | 0.0259 (0.0000) | 0.0209 (0.0000) |
| EARTH | 0.0243 (0.0093) | 0.0588 (0.0115) | 0.1506 (0.0017) | 0.0259 (0.0185) | 0.0219 (0.0003) |
| **EpiNode (Ours)** | **0.0023** (0.0005) | **0.0210** (0.0027) | **0.0195** (0.0002) | **0.0107** (0.0004) | **0.0090** (0.0002) |

*Note: ARIMA is deterministic and therefore no standard deviation is reported.*

data-driven models. Figure 9 and Figure A3 show the overall RMSE across both synthetic and real datasets.

### 4.5.2. PEAK ERRORS

On ILI (HHS 4) (Figure 10), EpiNode's predictions concentrate closest to the origin in both peak-timing and peak-magnitude error, achieving a timing bias of 0.0 weeks (zero seed-to-seed variance) and a peak-magnitude bias of $-0.017$ ($\sigma \approx 2 \times 10^{-3}$). The remaining methods split into two regimes. The first is low-variance but heavily biased predictors (ARIMA $+26$ weeks, LatentODE $-21$ weeks, EARTH $-15$ weeks, EINN-SEIRm $-12$ weeks). The second is approximately unbiased but high-variance predictors whose timing standard deviations span 9–19 weeks (LSTM $\pm9$, TimeKAN $\pm13$, EINN-SIRS $\pm18$, NeuralODE $\pm19$).

### 4.6. Applications

#### 4.6.1. PARAMETER INFERENCE

On synthetic data with ground truth for both unobserved compartments and time-varying parameters (Figures 11, A5, A6), EpiNode recovers both the latent compartment trajectories and the rate functions. The inferred susceptible and recovered populations follow the true $S(t)$ and $R(t)$, and the estimated $\beta(t)$, $\gamma(t)$, and $\delta(t)$ track their ground-truth values. EpiNode therefore reconstructs the full epidemic state and underlying parameters under partial observability, even when only infection counts are observed.

Figure A7 shows inferred parameters using the full observation window. For a representative region (HHS 4, Figure A7a), the estimated $\beta(t)$ exhibits clear seasonal oscillations aligned with major ILI waves, while $\gamma(t)$ remains stable and $\delta(t)$ varies smoothly at lower amplitude. Aggregated across all HHS regions (Figure A7b), $\beta(t)$ shows consistent seasonal modulation with moderate regional het-

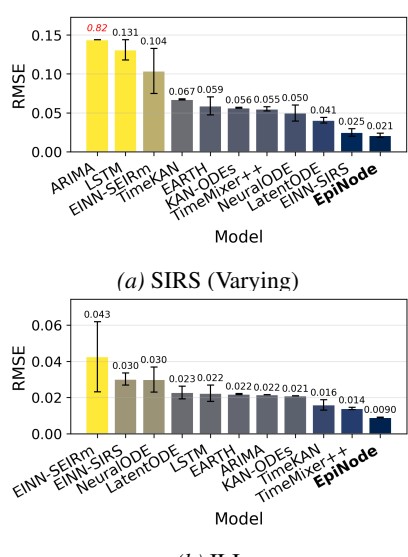

*(a)* SIRS (Varying)

*(b)* ILI

*Figure 9.* Overall benchmark RMSE on time-varying datasets.

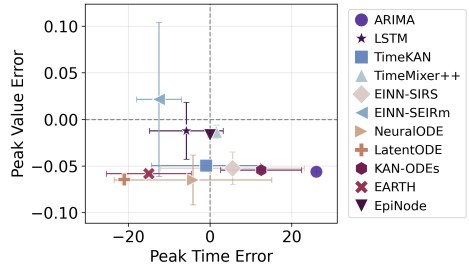

*Figure 10.* Peak error (magnitude & timing) of ILI

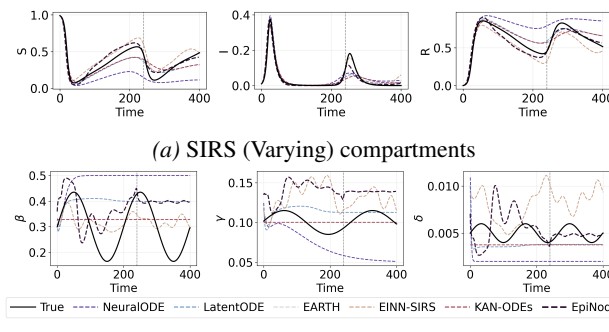

*(a)* SIRS (Varying) compartments

*(b)* SIRS (Varying) parameters

*Figure 11.* Unobserved compartments (a) and parameters (b) inferred from the full observation window.

erogeneity, reflecting shared seasonal forcing and region-specific transmission intensity. The inferred parameters remain smooth, bounded, and temporally coherent across regions, supporting stable and interpretable parameter recovery from real-world surveillance data.

## 5. Conclusion

This work highlights the importance of integrating multi-scale structure, continuous-time latent dynamics, and mechanistic constraints for epidemic forecasting under partial observability. Epidemic observations entangle multiple temporal effects, including long-term structural evolution, seasonal forcing, and short-term fluctuations within a single time series, making latent dynamics and time-varying parameters difficult to identify under partial observability. Multi-scale signal decomposition is a well-established paradigm in classical statistics and has become the dominant inductive bias in state-of-the-art deep learning forecasters (Huang et al., 2025; Zhang et al., 2022; Ouyang et al., 2021). For epidemic time series, we choose three modes correspond to known mechanistic drivers trend, season, residual, respectively. Our experimental results demonstrate that EpiNode

produces interpretable time-varying parameter trajectories and generates forecasts consistent with known behavior. These parameter estimates provide insights beyond point forecasts, enabling retrospective analysis and hypothesis generation about underlying drivers of epidemic dynamics.

On multiwave data (e.g., ILI), the empirical curve sometimes shows a double peak that EpiNode smooths into a single broad peak. The limitation lies in the mechanistic layer rather than the decomposition or latent dynamics. A single-strain SIRS cannot produce two well-separated peaks driven by independently circulating strains. Merging CDC ILINet with NREVSS subtype data (Kandula et al., 2017) confirms that the observed double peaks come from sequential waves of distinct influenza A subtypes (H3N2 followed by H1N1). Because the mechanistic layer is modular, this is addressable by swapping SIRS for multi-strain SIRS with cross-immunity (Andreasen et al., 1997), SEIRS for diseases with non-negligible incubation (Hethcote, 2000), each layer pluggable without changing the TSR decomposition or the latent Neural ODEs.

Under $I$-only supervision, infections identify the effective reproduction number $R_{\text{eff}}(t) = \beta(t)S(t)/\gamma(t)$ more sharply than they identify $\beta(t)$ and $\gamma(t)$ individually, a well-known feature of SIR-type likelihoods (Finkenstädt & Grenfell, 2005), and the bounded parameterization controls this only weakly. EpiNode's $\beta(t)$ trajectory is reliable because it co-varies with seasonal forcing, while $\gamma(t)$ should be read as the model's preferred decomposition of $R_{\text{eff}}$ rather than a direct recovery-rate estimate.

The current formulation focuses on single-region, deterministic dynamics and does not explicitly model uncertainty, spatial coupling, or intervention effects. Moreover, TSR decomposition is treated as a preprocessing step rather than a learned component, and performance may depend on the choice of decomposition method. Addressing these limitations by incorporating probabilistic latent dynamics, spatial interactions, or learnable decomposition modules represent promising directions for future work.

## Acknowledgements

We thank the ICML 2026 reviewers and the area chair for their thoughtful and constructive feedback, which materially improved the presentation of results.

This work is supported in part by US National Science Foundation grants CCF-1918770, IIS-2509636, IIS-2312794, and DBI-2412389. Any opinions, findings, and conclusions or recommendations expressed in this material are those of the authors and do not necessarily reflect the views of the sponsors.

## Impact Statement

This work develops EpiNode, a hybrid neural–physical framework for epidemic dynamics forecasting under partial observability that integrates multi-scale signal decomposition, controlled neural ODEs, and mechanistic SIRS dynamics within a unified architecture. By jointly forecasting infection trajectories and inferring bounded, interpretable time-varying epidemiological parameters, EpiNode bridges predictive performance and epidemiological insight, and generalizes across seasonal and non-seasonal diseases as well as single- and multi-wave regimes. Such accurate and interpretable forecasts have the potential to support public-health planning, early warning, and retrospective analysis by improving understanding of disease transmission and peak behavior.

At the same time, forecasts and inferred parameters derived from such models should be interpreted with care, as they depend on data quality, modeling assumptions, and incomplete observations. Misuse or overreliance on model outputs without appropriate domain expertise could lead to misguided decisions. We emphasize that EpiNode is intended as a decision-support and analysis tool rather than a standalone policy-making system. We do not foresee any novel ethical concerns beyond those commonly associated with applying machine learning to public-health data, and we encourage responsible use in conjunction with epidemiological expertise and transparent communication of model limitations and uncertainty.

## Software and Data

We release the full implementation at https://github.com/yiqisu/EpiNode.git.

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

## A. Failure Modes

We describe some of the architectures used for the experiments described in Section 2.

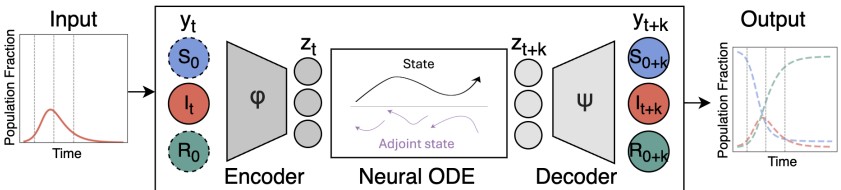

*Figure A1.* Overview of bidirectional AE-NODE framework.

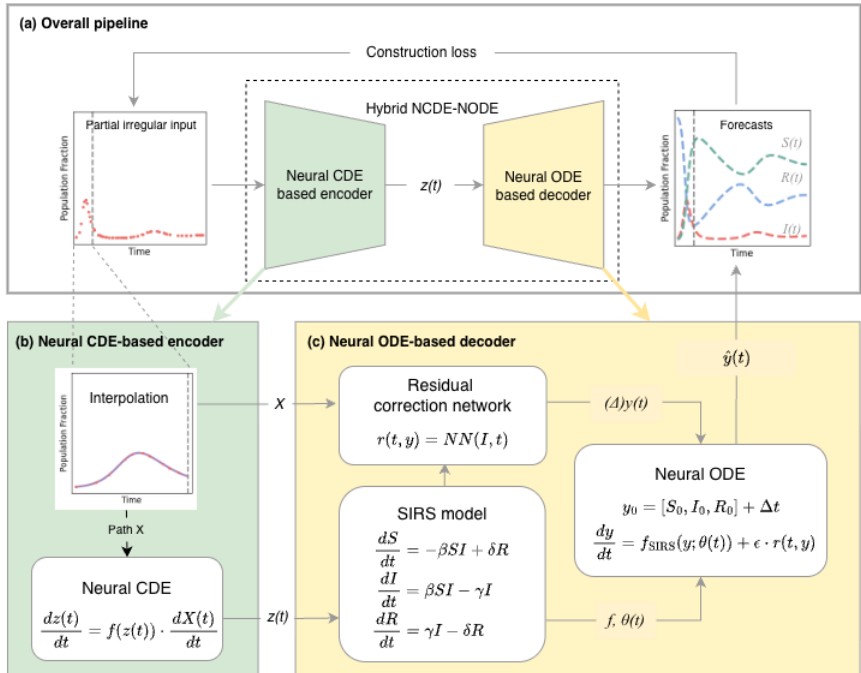

*Figure A2.* Overview of Neural CDE-ODE pipeline.

## B. Training and Inference Procedure

Algorithm 1 summarizes the end-to-end training and inference procedure for EpiNode. Given a partially observed infection time series, the algorithm first performs trend–season–residuals decomposition to extract multi-scale control signals, which optionally undergo time-delay embedding. These controls drive a set of latent continuous-time Neural ODEs whose states are fused to infer time-varying epidemiological parameters. The inferred parameters are then used to advance the mechanistic SIRS model via numerical integration, producing epidemic state forecasts. Model parameters are optimized by minimizing the prediction error on observed infections over the training window, with gradients propagated through both the Neural ODE solvers and the mechanistic dynamics. At inference time, the learned model is rolled out beyond the observation window to generate long-horizon forecasts, peak estimates, and interpretable parameter trajectories.

## C. Signal Decomposition Methods

We compare several signal decomposition techniques that differ in their assumptions about temporal structure, frequency separation, and modeling capacity. Each method decomposes the observed infection time series into components that are subsequently used to control latent continuous-time dynamics.

---

**Algorithm 1** EpiNode training and forecasting

---

**Require:** Observations $\{(t_i, I_i)\}_{i=0}^{T-1}$, split index $t_{\text{split}}$, delay params $(m, \tau)$, trend-fit window $L_{\text{fit}}$, step size schedule $\Delta t_i = t_{i+1} - t_i$; model components: VMD (§3), channel-aware extrapolation rules (9)–(11), collaborated ODE fields $f_{\theta_c}$ (14), fusion MLP (15), parameter decoder $g_\phi$ (17), SIRS solver (18)

**Ensure:** Forecasted states $\{\hat{\mathbf{y}}(t_i)\}_{i=0}^{T-1}$, parameters $\{\beta(t_i), \gamma(t_i), \delta(t_i)\}_{i=0}^{T-1}$

1: **Causal TSR decomposition:** $\{T_i, S_i, R_i\}_{i=0}^{t_{\text{split}}-1} \leftarrow \text{VMDDecompose}\big(\{I_i\}_{i=0}^{t_{\text{split}}-1}\big)$          (8)

2: **Channel-aware extrapolation.** Estimate $(a, b) \leftarrow \arg\min_{a,b} \sum_{j=t_{\text{split}}-L_{\text{fit}}}^{t_{\text{split}}-1} (T_j - a\,t_j - b)^2$ and $p_S \leftarrow$ dominant period of $\{S_j\}_{j=0}^{t_{\text{split}}-1}$ via periodogram (fallback 52 weekly, 7 daily). For each $i \in [t_{\text{split}}, T)$:

3:      $T_i \leftarrow a\,t_i + b$          (9)

4:      $k_i \leftarrow \lceil (i - t_{\text{split}} + 1)/p_S \rceil$; $S_i \leftarrow S_{i - p_S k_i}$          (10)

5:      $R_i \leftarrow 0$          (11)

6: Concatenate causal VMD output and extrapolated extensions to obtain $\{T_i, S_i, R_i\}_{i=0}^{T-1}$ on the full window.

7: **Time delay:** construct $\mathbf{u}_i^{(T)}, \mathbf{u}_i^{(S)}, \mathbf{u}_i^{(R)}$ from $\{T_i, S_i, R_i\}_{i=0}^{T-1}$ via delay embedding          (12)

8: **Initialize:** $\hat{\mathbf{y}}(t_0) = [1 - I_0,\ I_0,\ 0]^\top$          (1),(2)

9: $\mathbf{h}^{(T)}(t_0) = \mathbf{0}, \mathbf{h}^{(S)}(t_0) = \mathbf{0}, \mathbf{h}^{(R)}(t_0) = \mathbf{0}$          (13)

10: **for** epoch $= 1$ to $E$ **do**

11:      Set loss $\mathcal{L} \leftarrow 0$

12:      **for** $i = 0$ to $T - 1$ **do**

13:          **Latent ODE step (if $i > 0$):**

14:          **if** $i > 0$ **then**

15:              $\mathbf{h}^{(T)}(t_i) \leftarrow \text{ODEINT}(f_{\theta_T}, \mathbf{h}^{(T)}(t_{i-1}), [t_{i-1}, t_i]; \mathbf{u}_{i-1}^{(T)})$          (14)

16:              $\mathbf{h}^{(S)}(t_i) \leftarrow \text{ODEINT}(f_{\theta_S}, \mathbf{h}^{(S)}(t_{i-1}), [t_{i-1}, t_i]; \mathbf{u}_{i-1}^{(S)})$          (14)

17:              $\mathbf{h}^{(R)}(t_i) \leftarrow \text{ODEINT}(f_{\theta_R}, \mathbf{h}^{(R)}(t_{i-1}), [t_{i-1}, t_i]; \mathbf{u}_{i-1}^{(R)})$          (14)

18:          **end if**

19:          **Fuse latents:** $\mathbf{h}(t_i) \leftarrow \text{Fuse}([\mathbf{h}^{(T)}(t_i); \mathbf{h}^{(S)}(t_i); \mathbf{h}^{(R)}(t_i)])$          (15)

20:          **Decode rates:** $(\tilde{\beta}, \tilde{\gamma}, \tilde{\delta}) \leftarrow g_\phi(\mathbf{h}(t_i))$          (17)

21:          $\beta(t_i), \gamma(t_i), \delta(t_i) \leftarrow$ affine scaling of $(\tilde{\beta}, \tilde{\gamma}, \tilde{\delta})$          (17)

22:          **Physics rollout:** $\hat{\mathbf{y}}(t_{i+1}) \leftarrow \text{RK4Step}(\hat{\mathbf{y}}(t_i), \beta(t_i), \gamma(t_i), \delta(t_i), \Delta t_i)$          (18)

23:          Apply simplex stabilization / mass correction          (19)

24:          **if** $i < t_{\text{split}}$ **then**

25:              $\hat{I}(t_i) \leftarrow (\hat{\mathbf{y}}(t_i))_2$

26:              $\mathcal{L} \leftarrow \mathcal{L} + w_i(\hat{I}(t_i) - I_i)^2$          (20)

27:          **end if**

28:      **end for**

29:      $\mathcal{L} \leftarrow \mathcal{L}/t_{\text{split}}$          (20)

30:      Update parameters of $\{f_{\theta_c}\}_{c \in \{T,S,R\}}$, Fuse, $g_\phi$ by backpropagation through ODE solvers

31: **end for**

32: **return** $\{\hat{\mathbf{y}}(t_i)\}_{i=0}^{T-1}$ and $\{\beta(t_i), \gamma(t_i), \delta(t_i)\}_{i=0}^{T-1}$

---

**Moving average (MA).** The moving average decomposition applies a sliding-window average to smooth short-term fluctuations and extract a low-frequency trend component. Residuals are obtained by subtracting the smoothed signal from the original series. MA provides a simple baseline that captures coarse trends but does not explicitly model seasonality or frequency structure, and is sensitive to window size selection.

**Seasonal–trend decomposition using Loess (STL).** STL decomposes a time series into additive trend, seasonal, and residual components using locally weighted regression (LOESS). It assumes a fixed seasonal period and smooth temporal evolution, making it effective for stationary or weakly nonstationary seasonal patterns. However, STL does not enforce explicit frequency separation and may struggle when seasonal dynamics vary over time or across epidemic waves.

**Variational mode decomposition (VMD).** VMD decomposes a signal into a predefined number of intrinsic mode functions, each constrained to be band-limited around a learned center frequency. The decomposition is obtained by solving a variational optimization problem in the frequency domain that jointly minimizes bandwidth and reconstruction error. VMD produces components with well-separated frequency content and is robust to noise, making it well suited for isolating multi-scale epidemic dynamics.

**Wavelet.** Wavelet-based decomposition represents the signal using a set of scaled and shifted wavelet basis functions, yielding a multi-resolution time–frequency representation. This approach captures both local temporal variations and global structure. While wavelets provide strong localization in time and frequency, the resulting components are not necessarily narrowband, and performance depends on the choice of wavelet family and decomposition depth.

**SSA–VMD (hybrid decomposition).** SSA–VMD combines Singular Spectrum Analysis (SSA) with VMD to leverage the strengths of both methods. SSA first separates the signal into dominant subspaces corresponding to trend and oscillatory modes using low-rank trajectory matrices. VMD is then applied to selected components to further refine frequency separation. This hybrid approach improves robustness in noisy settings but introduces additional complexity and hyperparameters.

**Neural Koopman.** Neural Koopman methods learn a latent linear dynamical system by lifting the observed time series into a higher-dimensional feature space using neural networks. Decomposition is achieved by projecting dynamics onto learned Koopman modes and eigenfunctions. This approach is fully data-driven and expressive, but does not explicitly enforce band-limited or frequency-localized components, which can lead to entangled modes under strong nonstationarity.

## D. Benchmarking Models

- Autoregressive integrated moving-average (ARIMA) models serve as a classical statistical baseline for epidemic forecasting. ARIMA captures linear temporal dependencies through autoregressive and moving-average components and is commonly used for short-term epidemic prediction under near-stationary conditions.

- We include standard recurrent neural networks (RNN) and gated variants (LSTM/GRU) as nonlinear sequence modeling baselines. RNN-based models can capture complex temporal dependencies from data but operate as black-box predictors and do not enforce epidemiological constraints.

- TimeKAN (Huang et al., 2025) and TimeMixer++ (Wang et al., 2025) represent recent state-of-the-art univariate forecasters that leverage decomposition or multiscale mixing to capture long-range temporal patterns, providing strong data-driven baselines.

- To incorporate epidemiological structure, we include EINN (Rodríguez et al., 2023), a physics-informed neural model that embeds epidemic priors into neural forecasting without explicit continuous-time latent dynamics. As EINN employs SEIRm physics, we adapt the framework in two ways: 1) replace SEIRm with SIRS, 2) predict I rather than m. In both options, I is the only observed compartment.

- We further evaluate ODE-based continuous-time models, including Neural ODE (Chen et al., 2018), Latent ODE (Rubanova et al., 2019), and KAN-ODE (Koenig et al., 2024), which model temporal evolution via learned differential equations but do not enforce epidemic physics by default.

- Finally, we consider EARTH (Wan et al., 2025), a graph-based neural ODE model that integrates epidemiological dynamics with spatial coupling, evaluated only in multi-region settings where adjacency information is available. Given our focus on single region forecasting, we treat the region as a graph with $N = 1$ node and set adjacency $A = [1]$.

Table A1 summarizes the modeling capabilities of representative baselines and highlights the gaps that motivate our design.

*Table A1.* Comparison of modeling capabilities across baselines.

| Model | Univariate | Cont.-Time | Physics | TV Params | Interpretable | Non-Seasonality | Seasonality | Stable Long-Horizon |
|---|---|---|---|---|---|---|---|---|
| ARIMA | ✓ | ✗ | ✗ | ✗ | ✗ | ✗ | ✗ | ✗ |
| RNN (LSTM) | ✓ | ✗ | ✗ | ✗ | ✗ | ✗ | ✗ | ✗ |
| TimeKAN | ✓ | ✗ | ✗ | ✗ | ✗ | ✗ | ✓ | ✗ |
| TimeMixer++ | ✓ | ✗ | ✗ | ✗ | ✗ | ✗ | ✓ | ✗ |
| EINN | ✓ | ✗ | ✓ | ✗ | ✓ | ✗ | ✗ | ✗ |
| Neural ODE | ✓ | ✓ | ✗ | ✗ | ✗ | ✗ | ✗ | ✗ |
| Latent ODE | ✓ | ✓ | ✗ | ✗ | ✗ | ✗ | ✗ | ✗ |
| KAN-ODE | ✓ | ✓ | ✗ | ✗ | ✗ | ✗ | ✗ | ✗ |
| EARTH | † | ✓ | ✓ | ✓ | ✓ | ✓ | ✓ | ✓ |
| EpiNode (ours) | ✓ | ✓ | ✓ | ✓ | ✓ | ✓ | ✓ | ✓ |

*"Univariate" indicates native support for single-series forecasting. "TV Params" indicates inference of time-varying parameters. "Stable Long-Horizon" reflects empirically observed robustness under long rollout. † EARTH is designed for multi-region forecasting; the univariate case reduces to a degenerate one-node graph without spatial coupling.*

## E. Dataset Details

To comprehensively evaluate epidemic forecasting performance under partial observability, time-varying dynamics, and physics mismatch, we benchmark our approach on a diverse suite of synthetic and real-world datasets, spanning multiple disease regimes, observation settings, and experimental conditions.

### E.1. Synthetic Datasets

**SIRS with time-fixed and time-varying parameters.** We generate synthetic epidemics using the SIRS model under multiple parameter regimes. In the fixed setting, transmission, recovery, and immunity-loss rates remain constant over time, serving as a baseline for identifiability under stationarity. In the periodic setting, parameters vary smoothly and periodically to emulate seasonal forcing, capturing recurring epidemic patterns.

**Different epidemic physics.** To assess robustness to physics mismatch, we additionally simulate data from alternative compartmental models. The SIR setting removes immunity waning, testing the model's ability to adapt when the assumed SIRS structure is overparameterized. The SEIRS setting introduces an exposed compartment, increasing latent-state complexity and evaluating performance when the true dynamics deviate from the assumed model class.

### E.2. Real-World Datasets

Influenza-like illness (ILI) surveillance data (https://gis.cdc.gov/grasp/fluview/fluportaldashboard.html) provides a canonical example of strongly seasonal epidemic dynamics. We consider two settings: single-wave segments, obtained by isolating individual seasonal outbreaks, and multi-wave sequences spanning multiple years. This allows evaluation of both early outbreak forecasting and long-term seasonal recurrence. We evaluate across multiple U.S. Department of Health and Human Services (HHS) regions (HHS 1–10, covering the entire continental United States) and across multiple time periods, capturing heterogeneity in epidemic progression and reporting practices.

## F. Choice of Train–Test Splits

EpiNode follows a principled, regime-aware evaluation design tailored to epidemic dynamics, with split placement determined by the underlying wave structure of each dataset.

**Early-stage splits for single-peak dynamics.** For single-wave epidemics, the realistic forecasting task is early-stage prediction. We use small training fractions (e.g., $0.2$ for SIR) so that the model is required to extrapolate the epidemic

trajectory *through and beyond* the peak rather than merely interpolating between observed peaks.

**Cycle-aware splits for multi-peak dynamics.**    For multi-wave epidemics the goal is to forecast future waves from historical cycles. We place $t_{\text{split}}$ after observing one or more prior cycles, typically near the start of a new wave to be forecasted (e.g., $0.6$ for SIRS (Varying), $0.7$ for ILI), so the model must generalize from prior cycles to subsequent ones.

**Robustness to split placement.**    To verify that the reported gains do not depend on favorable split placement, we evaluate every dataset at multiple splits spanning pre-, around-, and post-peak locations (Table A2). All reported metrics in the main paper and appendix are aggregated across these splits and seeds.

*Table A2.* Train–test splits evaluated per dataset, with the forecast regime each split probes.

| Dataset | Splits | Regime probed |
|---|---|---|
| SIRS (Fixed) | 0.3, 0.5, 0.6 | pre-peak, around-peak, post-peak |
| SIRS (Varying) | 0.5, 0.6 | wave-1 end, mid wave-2 |
| SIR (Fixed) | 0.1, 0.2 | early-outbreak, growth phase |
| SEIRS (Fixed) | 0.5, 0.6 | around-peak, post-peak |
| ILI (weekly) | 0.7, 0.8, 0.9 | growth, peak, decay |

## G. Implementation via Bounded Parameterization.

Rather than fixing parameters or imposing strong smoothness priors, we implement these ranges through a bounded neural parameterization, where raw network outputs are mapped via a sigmoid function and affine scaling into the prescribed intervals. This approach follows prior work on constrained neural modeling of dynamical systems and epidemic processes (Raissi et al., 2019; Chen et al., 2018). By encoding epidemiological knowledge as soft constraints, the model remains expressive while producing parameter trajectories that are interpretable, numerically stable, and consistent with known disease characteristics.

## H. Supplementary Results

### H.1. Performance Evaluation

#### H.1.1. FORECASTING ACCURACY

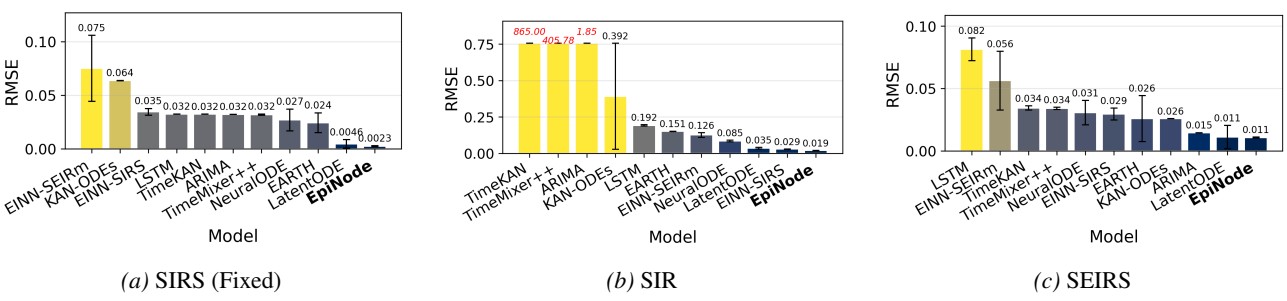

*(a)* SIRS (Fixed)          *(b)* SIR          *(c)* SEIRS

*Figure A3.* Overall benchmark RMSE on time-fixed datasets.

Figure A3 complements the real-data benchmark (Figure 9) with overall RMSE on the three time-fixed synthetic datasets (SIRS, SIR, SEIRS). EpiNode achieves the lowest mean RMSE on all three datasets. The win on SIRS (Fixed) shows that the model is accurate when the assumed physics matches the data, and the wins on SIR (Fixed) and SEIRS (Fixed) show that the latent control signals absorb the residual structure introduced by the mismatch, so the advantage extends from non-stationary settings highlighted in the main paper to stationary regimes with both matched and mismatched physics. Univariate Transformer-style forecasters (TimeKAN, TimeMixer++) are off-scale on SIR (Fixed).

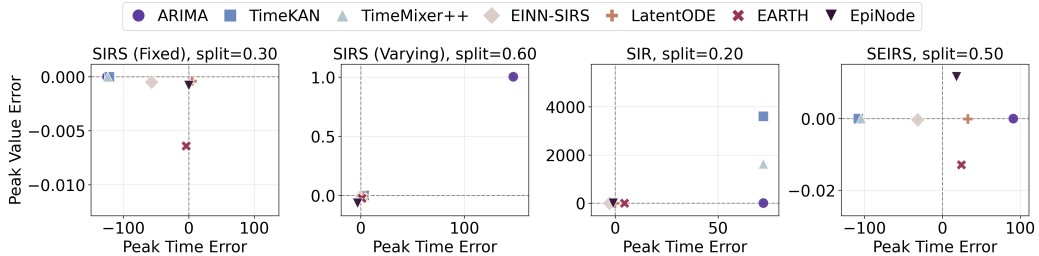

*Figure A4.* Peak error (magnitude & timing) across synthetic datasets

*Table A3.* Peak timing error across synthetic and real datasets (mean ± std).

| Method | SIRS (Fixed) | SIRS (Varying) | SIR (Fixed) | SEIRS (Fixed) | ILI |
|---|---|---|---|---|---|
| ARIMA | -125 | 147 | 72 | 91 | 26 |
| LSTM | -125.0 (0.0) | 0.6 (0.5) | 28.8 (1.3) | -33.2 (4.8) | -5.8 (9.0) |
| TimeKAN | -121.4 (1.6) | 3.4 (6.4) | 72.0 (0.0) | -107.8 (1.9) | -1.0 (13.3) |
| TimeMixer++ | -123.8 (1.5) | 2.0 (5.8) | 72.0 (0.0) | -104.2 (2.6) | 1.5 (0.5) |
| EINN-SIRS | -57.0 (3.2) | 0.8 (1.7) | -2.8 (0.4) | -31.6 (49.3) | 5.5 (17.6) |
| EINN-SEIRm | 98.2 (63.1) | 93.8 (43.6) | 6.2 (25.9) | -39.6 (26.8) | -12.5 (5.5) |
| NeuralODE | -47.6 (109.4) | 10.0 (13.7) | -5.8 (0.4) | 51.0 (80.0) | -15.0 (10.4) |
| LatentODE | 5.0 (11.7) | 3.0 (1.7) | **-0.2** (0.4) | 32.6 (29.7) | 12.5 (9.9) |
| KAN-ODEs | -61.3 (6.6) | -13.0 (0.0) | 17.5 (33.8) | -69.7 (12.8) | -21.0 (0.0) |
| EARTH | -4.0 (117.4) | 1.2 (1.1) | 4.5 (8.6) | 24.3 (52.0) | -4.2 (19.3) |
| **EpiNode (Ours)** | **0.0** (0.0) | **-0.5** (2.5) | -1.0 (0.0) | **19.5** (1.5) | **0.0** (0.0) |

*Note: ARIMA is deterministic and therefore no standard deviation is reported.*

### H.1.2. PEAK ERRORS

Figure A4 and Table A3 report peak time and peak value errors on the synthetic datasets. EpiNode has the smallest mean peak-timing error on four of five datasets. Many baselines achieve low pointwise RMSE but still mis-time the peak. Models relying on latent dynamics or static parameters accumulate small growth-rate errors that amplify near peaks, predicting them too early under over-aggressive growth or too late under excessive damping.

## H.2. Applications

### H.2.1. PARAMETER INFERENCE

On synthetic data with ground truth, EpiNode recovers the unobserved $S(t), R(t)$ trajectories (Figure A5) and the time-varying $\beta(t), \gamma(t), \delta(t)$ within their prescribed bounds (Figure A6). However, baselines drift, flatten, or produce no parameter outputs. On real ILI data across all ten CDC HHS regions (Figure A7), the inferred $\beta(t)$ is smooth, bounded, and shows the expected winter-peaking modulation, with consistent regional patterns and no per-region tuning.

### H.2.2. REGIONAL DYNAMICS

Figure A8 shows the forecasts across all ten HHS regions. In HHS 4 and HHS 8, the proposed model accurately captures both the timing and magnitude of the post-split infection peak, because these regions have regular seasonal structure and smoothly varying trends that the TSR decomposition cleanly separates. In contrast, forecasts in HHS 7 and HHS 10 are less accurate, particularly around the post-split surge. These regions exhibit more irregular dynamics and abrupt changes in peak amplitude that are weakly represented in the training window, producing larger residual components and a pronounced distribution shift at forecast time. The inferred transmission dynamics therefore underestimate the rapid increase and underpredict the epidemic peak. These findings show that forecast accuracy depends strongly on the structural regularity of regional epidemic dynamics.

## H.3. Ablation Studies

We conduct extensive ablations to isolate the contribution of key architectural components.

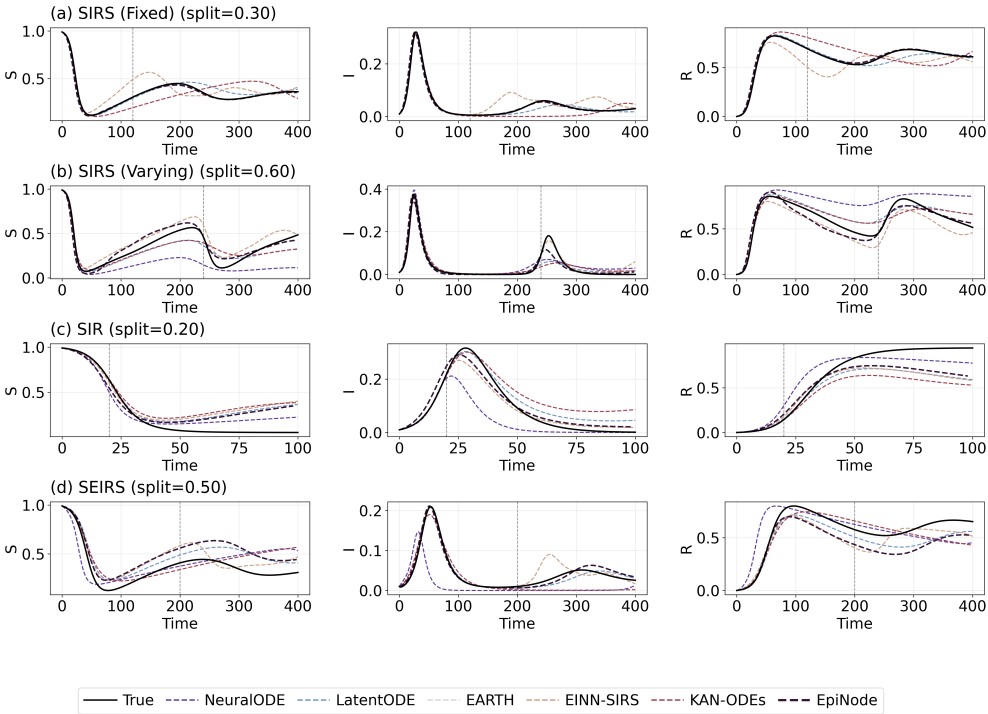

*Figure A5.* True and predicted compartments for synthetic datasets.

### H.3.1. SINGLE LATENT ODE VARIANT

Figure A9a compares architectures with a single latent ODE (1ODE) versus three collaborated latent ODEs (3ODEs) corresponding to trend, seasonal, and residual components. Under identical training conditions, 3ODEs consistently achieves lower error than 1ODE variants, particularly when time-delay embedding is enabled. This indicates that disentangling multi-scale dynamics into separate latent flows improves identifiability and long-horizon stability.

### H.3.2. NUMBER OF DECOMPOSED COMPONENTS

Figure A9b shows the comparison among 1 component (C) *vs* 2C *vs* 3C. Across all datasets, RMSE decreases with one to two to three components, suggesting that threecomponents provide the most effective signals to separate slow structural evolution from seasonal forcing.

### H.3.3. TSR DECOMPOSITION METHOD

Figure A9c shows that VMD-based decomposition ranks best for almost all datasets and the second for SEIRS model, likely due to its explicit frequency localization and robustness to noise.

### H.3.4. TIME-DELAY EMBEDDING

Incorporating time-delay embedding further improves accuracy by providing temporal context to each component (Figure A9a). An exception occurs in the SIRS setting with time-fixed parameters, where the dynamics are noise-free and stationary; in this case, time-delay embedding offers little benefit, as the system evolution is determined by the current state.

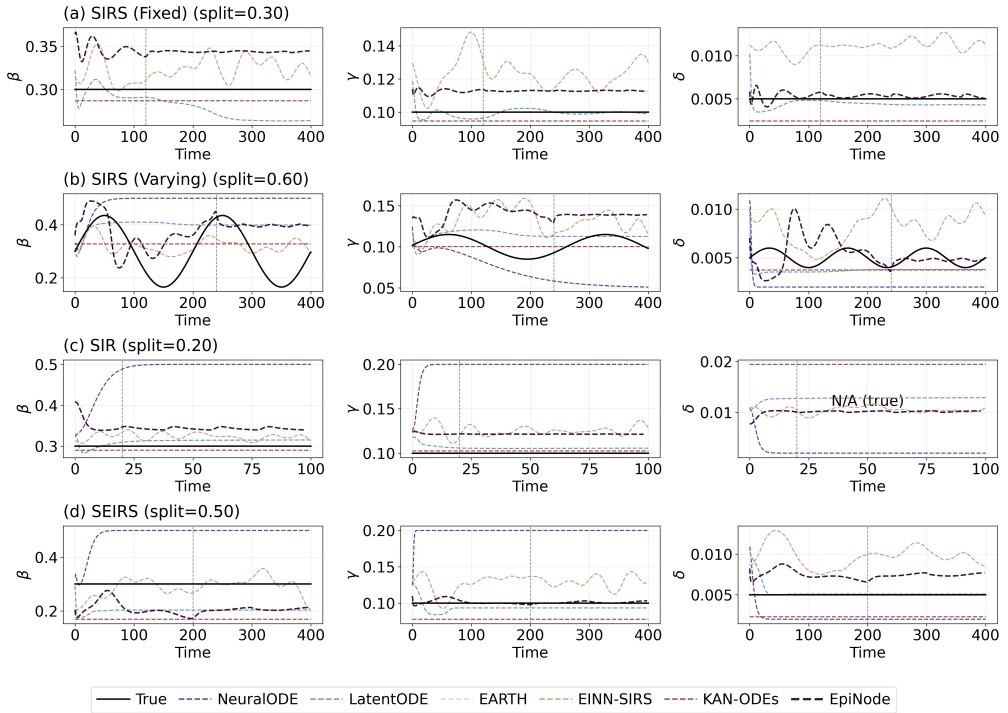

*Figure A6.* True and predicted parameters for synthetic datasets.

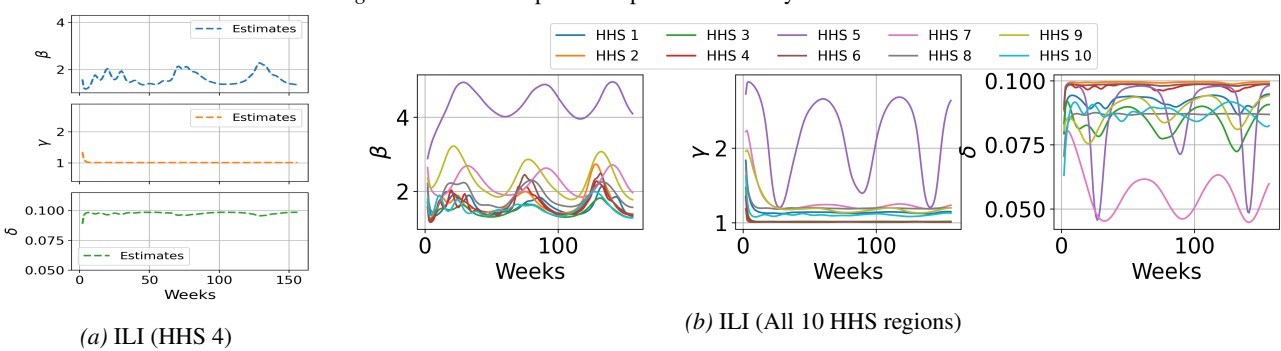

*(a)* ILI (HHS 4)

*(b)* ILI (All 10 HHS regions)

*Figure A7.* Inferred time-varying epidemiological parameters using the full observation window. (a) Estimated parameter rates for a representative HHS region (HHS 4). (b) Aggregated parameter estimates across all ten HHS regions.

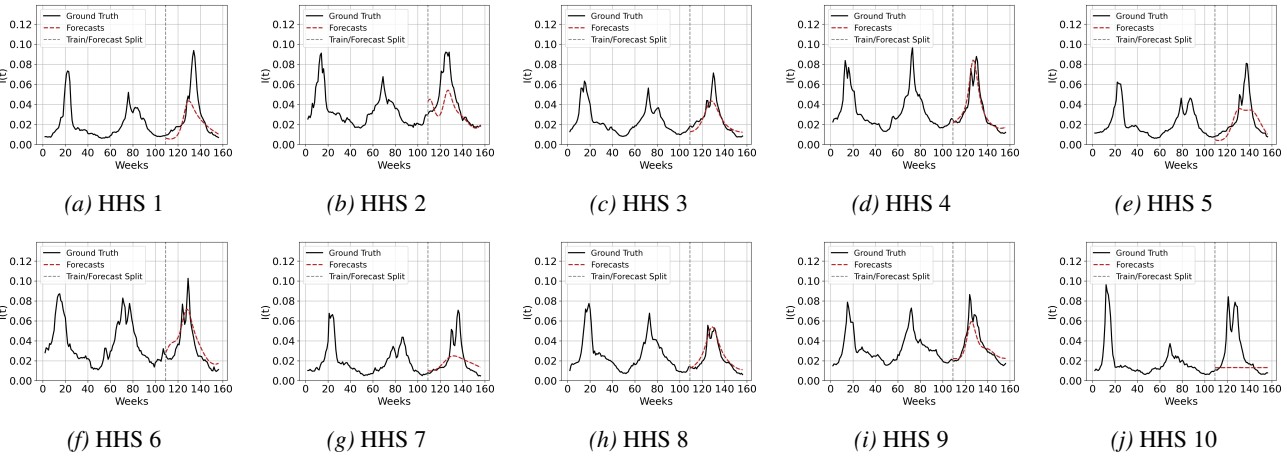

*(a)* HHS 1    *(b)* HHS 2    *(c)* HHS 3    *(d)* HHS 4    *(e)* HHS 5

*(f)* HHS 6    *(g)* HHS 7    *(h)* HHS 8    *(i)* HHS 9    *(j)* HHS 10

*Figure A8.* Structured regional dynamics enable accurate forecasting across all HHS regions at split $0.7$.

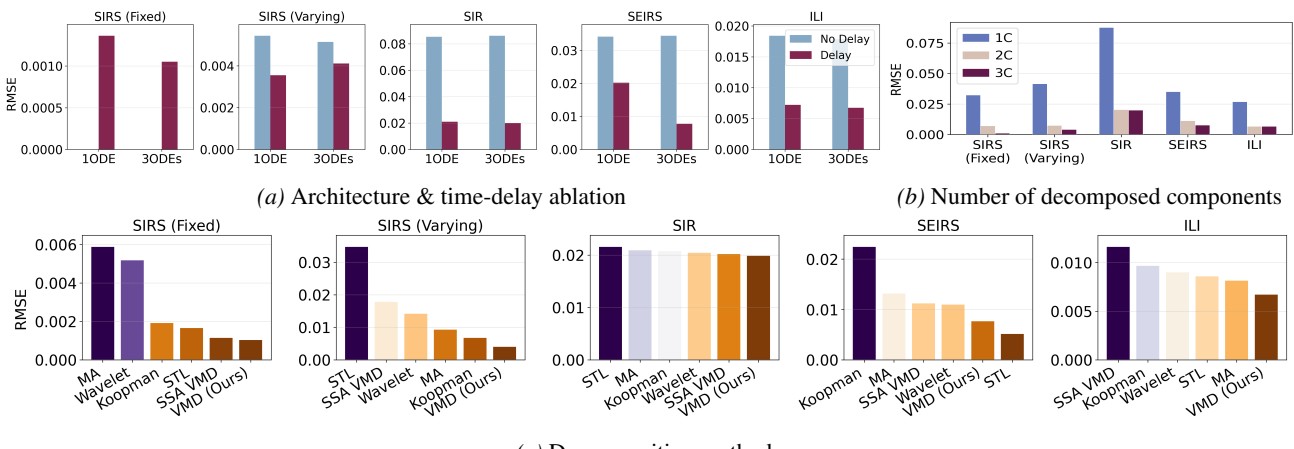

*(a)* Architecture & time-delay ablation        *(b)* Number of decomposed components

*(c)* Decomposition methods

*Figure A9.* Ablation results across datasets. (a) Comparison of 1ODE vs. 3ODEs with and without time-delay embedding. (b) Comparison of 1-component *vs* 2-components *vs* 3-component decomposition. (c) Comparison of signal decomposition methods.

