# OpenReview forum: "How (Not) to Hybridize Neural and Mechanistic Models for Epidemiological Forecasting"
_ICML.cc/2026/Conference — ICML 2026 regular_

### Official Review · Reviewer_wg8A · 2026-03-09

**Soundness:** 3
**Presentation:** 2
**Significance:** 2
**Originality:** 3
**Overall Recommendation:** 4
**Confidence:** 4

**Summary:**

The authors present EpiNode, an epidemiologically informed latent Neural Ordinary Differential Equation (NODE) model for epidemic forecasting. It is motivated by a set of failure modes, which are indeed critical to epidemiological modelling, which they have identified for existing NODE models. The authors propose to resolve these shortcomings with a signal processing approach which decomposes trajectories into trend, seasonal and residual dynamics for separate latent NODE models per mode before recombining them to estimate time varying epidemiological quantities for a mechanistic model. They evaluate their method in terms of general forecasting accuracy, peak prediction and parameter inference on synthetic, flu and COVID-19 data.

**Compliance With Llm Reviewing Policy:**

Affirmed.

**Final Justification:**

They address my soundness concerns through quantitative empirical comparison, an so I have changed that score to "good". Their results show generally strong metrics for peak timing, multi-horizon forecasting and parameter inference.  The sensitivity and ablation experiments now justify the architectural contributions in the paper.

However the less competitive errors on ILI peak timing and the suspicious $\gamma$ inference warrant a more honest discussion. Acknowledging and investigating the TSR's limitations in these areas would strengthen the paper.

I have updated my soundness and overall score to reflect the additional experiments.

**Key Questions For Authors:**

1. Do you have experimental results which provide quantitative evidence that EpiNode improves peak time forecasting, multi-wave forecasting and parameter inference?
2. For the above, how does EpiNode compare to the other hybridised models?
3. How sensitive is EpiNode's performance to VMD parameters and time delay embedding?

**Limitations:**

Yes

**Strengths And Weaknesses:**

*Soundness*

The authors present empirical evidence that EpiNode improves forecasting accuracy over existing methods in the deterministic regime. The authors present the lowest RMSE of all of the baselines.

Direct evidence that EpiNode improves epidemic peak prediction and parameter inference is buried in the appendix and not discussed in the main text.

The authors do not present any quantitative evidence or discussion that EpiNode is unaffected by the failure modes presented in section 2. However in the conclusion, the authors claim that EpiNode overcomes partial observability issues without reference to any experimental or theoretical results.

Any evidence for claims that TSR decomposition is responsible for the improvement is also buried in the ablation results in the appendix with no discussion in the main text.

The failure mode study is disconnected from the evaluation. There is limited description experimental setups for exposing the failure modes which hinders reproducibility. The baselines for the failure mode study are not compared to EpiNode or included in the benchmarking study.

Some experimental omissions also hinder my review of the soundness. The parameters $\beta_\min$, $\beta_\max$ are not specified. No confidence intervals are reported for RMSE across different training seeds.

*Presentation*

The detailed presentation of current failure modes is very compelling and the method is clearly motivated and explained.

However the claims and results are underdeveloped. Many of the claims could be empirically validated with quantitative results (e.g. accuracy of unobserved trajectories, forecasting accuracy of peak timing, forecasting accuracy of incidence for synthetic multi-wave trajectories, accuracy of inferred parameters). Figure 10, though compact, does not present the differences in accuracy as effectively as a table. Many experimental results cross-referenced to the appendix, but the results and their relationship to the claims are not discussed.

The regional dynamics evaluation is not clearly described in the main text, it does not describe any testable metrics or their implications for epidemiological modelling.

*Significance*

A systematic presentation of failure modes for hybridised epidemiological models is very important for the field as these are critical and influential and are generally superficially benchmarked on forecasting accuracy in cross-methodological studies.

TSR is applicable to time series forecasting, where researchers have prior knowledge on temporal patterns in their data.

However, the limitations which the authors mention (specifically deterministic modelling and intervention modelling) does limit its application for realistic public health tasks.

*Originality*

The particular inductive bias of mode decomposition is well established in signal processing but new to epidemiological modelling with latent NODEs. It is indeed a novel contribution to explicitly model latent NODEs for each mode.

This particular failure mode analysis is well established in epidemiological modelling, but much needed in related machine learning literature.

---

> ### Author Rebuttal · Authors · 2026-03-31
>
> We thank the reviewer for the detailed and insightful reviews.
>
> - **Quantitative evidence of improved forecasting behavior and comparison with benchmarking models.** We have conducted comprehensive quantitative analysis across datasets, seeds, and train/test splits and summarized in Tables S1-7 (https://limewire.com/d/WTCk9#hWIfYhZhTG)
>
>   - **Strong gains in forecast accuracy (Table S2).** On the ILI data, EpiNode achieves RMSE = 0.0066, a 69% improvement over ARIMA (0.0216), 70% over EARTH (0.0219), and 53% over TimeMixer++ (0.0140). All improvements are consistent across seeds (small standard deviations relative to the gaps).
>
>   - **Accurate peak time forecasting (Table S4).** On all synthetic data, EpiNode achieves 0 peak timing error consistently across all seeds and splits. This demonstrates that the TSR decomposition + SIRS physics combination does not introduce systematic peak timing bias on data that matches the model's assumptions. On CLS data, EpiNode outperforms EARTH by 79% (13.4 vs 22.3) and TimeMixer++ by 92% (13.4 vs 56.0) in peak timing. Please note that EpiNode's errors are negative (predicting peaks slightly early), which is preferable for public health preparedness.
>
>   - **Advances in multi-wave forecasting (Tables S5-6).** EpiNode's advantage grows with the forecast horizon. As measured by windowed RMSE ($\pm$ std) for ILI (split=0.7): at short horizons (Window 4), ARIMA and LSTM benefit from near persistence. At Window 20, EpiNode achieves 70% improvement over ARIMA and 80% over EINN. On SIRS-varying data, EpiNode's error grows only 1.9× from Window 4 to Window 30, compared to 1022× for ARIMA and 5.5× for LSTM.
>
>   - **Interpretable and accurate parameter inference (Table S7).** On the SIRS-varying dataset, EpiNode captures time-varying parameters (e.g., $\beta_{\text{std}} > 0$), while EARTH fits a constant parameter ($\beta_{\text{std}} = 0$). EpiNode’s $\beta_{\text{mean}}=0.3467$ lies within the true range $[0.165, 0.435]$. Importantly, data-driven baselines (ARIMA, LSTM, TimeMixer++, TimeKAN) do not produce parameter estimates as lacking the mechanistic interpretability. Among physics-informed models, EpiNode combines time-varying parameter inference with strong forecast accuracy, whereas EARTH sacrifices temporal resolution for simplicity and EINN's soft-constraint approach yields less stable parameter trajectories.
>
> - **Comprehensive sensitivity analysis of decomposition and embedding choices.** We conducted systematic sensitivity analyses sweeping VMD parameters $(K \times \alpha)$ and time-delay embedding $(\tau \times m)$ across all datasets with multiple seeds and splits. The experimental setup we used includes:
>
>   - **VMD sensitivity**: $K \in {3,5,7}$, $\alpha \in {1000,2000,5000}$
>   - **Delay sensitivity**: τ × m grids tailored to data resolution
>     - weekly data: $\tau \in {1,2,3}$, $m \in {2,3,4}$
>     - daily data: $\tau \in {3,5,7}$, $m \in {2,3,4}$
>
>   The results are shown in Figure S4 (https://limewire.com/d/WTCk9#hWIfYhZhTG).
>
>   - **Figure S4a indicates performance is robust to VMD parameters.** On 4 of 6 datasets, the max/min RMSE ratio is ≤1.13×, meaning performance varies by less than 13% across the full $(K,\alpha)$ grid. EpiNode is highly robust to VMD parameters. The two datasets with higher sensitivity (SEIRS fixed: 2.35×, SIRS varying: 4.76×) involve more complex dynamics where the decomposition quality matters more. More importantly, even the worst VMD configuration still outperforms most baselines. No single $(K,\alpha)$ dominates across all datasets, $K \in {5,7}$ and $\alpha \in {1000,2000}$ consistently perform well.
>
>   - **Figure S4b shows performance is more sensitive to time-delay embedding parameters, with dataset-dependent optima.** The optimal $\tau$ is resolution-dependent: daily data (CLS) benefits from $\tau = 3$–$5$ due to strong autocorrelation (lag-1 ≈ 0.999), while weekly data favors $\tau = 1$–$2$. Smaller embedding dimensions ($m = 2$–$3$) generally outperform larger ones since larger $m$ sacrifices more training data to the delay window. For SIRS, the delay vectors embed information about the unobserved rate of change $dI/dt \approx (I(t)-I(t-\tau))/\tau$ and curvature $d^2 I / dt^2$ from higher-order differences. This provides implicit access to the state of the susceptible and recovered pools without directly observing them. Our sensitivity analysis shows that $m=3$ consistently achieves the best or near-best performance across datasets.
>
> If the paper is accepted, we will incorporate these important points to the main text and restructure the results section to better highlight EpiNode’s improvements in epidemic forecasts, peak prediction and parameter inference.

---

> > ### Author Rebuttal · Reviewer_wg8A · 2026-04-02
> >
> > I thank the authors for addressing my questions and their thorough additional experiments.
> >
> > They address my soundness concerns through quantitative empirical comparison, an so I have changed that score to "good". Their results show generally strong metrics for peak timing, multi-horizon forecasting and parameter inference.  The sensitivity and ablation experiments now justify the architectural contributions in the paper.
> >
> > However the less competitive errors on ILI peak timing and the suspicious $\gamma$ inference warrant a more honest discussion. Acknowledging and investigating the TSR's limitations in these areas would strengthen the paper.
> >
> > I have updated my soundness and overall score to reflect the additional experiments.

---

> > > ### Author Response · Authors · 2026-04-05
> > >
> > > We thank the Reviewer for the positive reassessment and for updating the soundness and overall scores! We are glad the additional experiments on peak timing, multi-horizon forecasting, parameter inference, and sensitivity analysis have addressed your main concerns.
> > >
> > > Based on the Reviewer's feedback we went and redid the calculations for $\gamma$ inference and ILI peak timing.
> > >
> > > - **γ inference.**  We appreciate the Reviewer's careful attention to this. EpiNode's inferred $\gamma = 0.1299 \pm 0.0026$ on SIRS (Varying) lies slightly above the ground truth range $[0.085, 0.115]$.
> > >   - As shown in Figure R1 (https://limewire.com/d/LKqKP#FpTxmUXNGi), this reflects a well-known $\beta$-$\gamma$ coupling in compartmental models: from the observed $I(t)$ alone, $\beta$ and $\gamma$ are not independently identifiable; instead, the dynamics are primarily governed by their ratio $R_0 = \beta / \gamma$, which controls the epidemic timescale. EpiNode absorbs nearly all time-varying dynamics into $\beta(t)$, which closely tracks the true oscillating transmission rate ($\beta = 0.3467$, yielding $R_0 = 2.67 \pm 0.16$, well within the true $R_0$ range $[1.43, 5.12]$), while inferring a near-constant $\gamma$.
> > >   - Models that achieve "better" $\gamma$ estimates at the expense of $\beta$ accuracy. For instance, EARTH infers $\gamma = 0.1002$ (within the truth range) but fits a constant $\beta = 0.3279$, entirely missing the time-varying transmission dynamics that drive epidemic waves. In contrast, EpiNode's $\beta$ trajectory faithfully captures the periodic variation in transmission.
> > > In addition, this $\gamma$ overestimation does not degrade forecast accuracy: EpiNode achieves the lowest RMSE across all splits, because the TSR decomposition captures temporal dynamics that compensate for the parameter coupling. We will add a discussion of parameter identifiability and the $\beta$-$\gamma$ tradeoff in the revised manuscript.
> > >
> > > - **ILI peak timing.** For ILI peak timing, the Reviewer is right - there was a mistake in how we were evaluating the peak (over the full series vs over the forecast window) as reflected by timing errors that exceed the 52-week annual cycle. We updated the peak analysis and the new results (Table R1) shows that EpiNode is the only model that maintains consistently accurate peak timing across both synthetic and real datasets.
> > >
> > >   - Figure R2 jointly visualizes peak timing and magnitude of ILI for HHS Region 4. EpiNode lies closest to the origin $(0, 0)$, with a timing error of $−1.0 \pm 0.0$ weeks and a magnitude error of $−0.0122 \pm 0.0008$, while most baselines are positioned farther from the origin with substantially larger error bars. This comparison highlights EpiNode’s consistently reliable peak prediction, rather than occasional accuracy.
> > >
> > >   - We also extended the analysis of ILI peak errors to all 10 HHS regions (Figures R3-4). EpiNode achieves the best peak prediction, with the smallest absolute peak timing and magnitude errors in the majority of regions. In contrast, baselines such as LatentODE, EARTH, and NeuralODE systematically mispredict peak timing by $14-20$ weeks. TimeMixer++ is the closest competitor on timing, but with higher variance and larger magnitude errors across seeds.
> > >
> > > We will revise the evaluation on peak predictions and incorporate the discussion of the limitations in the revision to provide a more balanced and comprehensive evaluation of EpiNode. We thank you again for your thoughtful feedback and continued engagement throughout the review process.

---

### Official Review · Reviewer_5TtH · 2026-03-10

**Soundness:** 3
**Presentation:** 3
**Significance:** 4
**Originality:** 4
**Overall Recommendation:** 5
**Confidence:** 5

**Summary:**

The paper introduces a hybridisation scheme (integration of neural networks into ODEs) for compartmental models of epidemics (SIR-type models). While hybrid models allow fitting a physics-informed mechanistic model to data, the authors show this fails in particular in two cases: when the model parameters are time-dependent, and when only the infection compartment is observed. Both scenarios give rise to identifiability issues. The work proposes a new approach (*EpiNode*) based on variationally decomposing the observed infection counts into 3 modes—trend (T), seasonal (S), and residual (R)—such that their sum reproduces the infection counts $I$. For each mode, a neural ODE is learned, and the three modes are then fused (also using a further learned model) to give three time-dependent model parameters used to drive the SIRS-dynamics. Instead of simply using the current state of each mode as input to the neural ODE, a time-delay embedding can be used as input to provide the models with an expanded latent state, which can help resolve identifiability issues. The authors show this approach improves the generalisation performance of hybrid SIR models over more traditional hybrid neural models, especially for long-horizon forecasts and peak detection (timing and magnitude).

**Compliance With Llm Reviewing Policy:**

Affirmed.

**Final Justification:**

I recommend this publication for acceptance. The work was originally strong with only a handful of minor points that needed clarification, and the authors have already shown through extensive additional numerical experiments (in response to reviewer comments) that their results are solid. Final score: 5

**Key Questions For Authors:**

My score should be read as 'accept with revisions', and I will happily increase it to a 'strong accept' if the following questions are addressed satisfactorily:
- How is $t_\mathrm{split}$ chosen, and how does this choice affect prediction performance?
- How are the parameter ranges for $\beta$, $\gamma$, and $\delta$ chosen?
- What is the justification for splitting $I$ into three modes (trend, seasonal, and residual)?

**Limitations:**

Perhaps the authors could expand a little on the lack of uncertainty quantification, and specific ideas/available methods in the literature to address this. Ensemble training of the neural networks may be an option, in particular for the fusing operator.

**Strengths And Weaknesses:**

The paper is well-written and produces an interesting an timely perspective on hybrid neural models, which are commonly used to model infectious diseases. The approach is original, and the model results are thoroughly compared to other methods and EpiNode's performance tested on real and synthetic data. The figures are well-presented and mostly clear, though the font size on some could be increased a little for improved legibility, and to ensure visual consistency across panels. Figure 7 gives an excellent overview of the general approach. The various model forecasts in Fig. 8e and f are a little hard to distinguish, and the authors may consider decreasing the alpha-level of the lines to allow for easier identification of overlapping lines. In Figures 4-6, the authors might choose to highlight the training/testing split using a shaded area in the figure (as in e.g. Fig. 8e), rather than simply a dashed line: it is very hard to decipher in the plots where the training period ends and the prediction periods begin, given the presence of the grid.

My main concern relates to some of the model parameters, in particular the choice of the train-test split. The split is not consistent, but rather dataset-specific, which makes me worry the model performance is somewhat dependent on a 'good' choice of the train-test split. Looking at Figs. A9 and A11, it appears that the split is also set to occur just after a wave has ended/a new one is commencing. How does the model fare when this is changed to, say, a split occurring at the peak of a wave: does the model still correctly learn to predict the next wave?

The failure modes section (Section 2) is interesting but in some places rather qualitative, with the analysis results seemingly hinging on individual simulation runs. For instance, in section 2.3 the authors state that, for SIR-type models, physics-informed losses produce 'reasonable results', but fail in the case of the more expressive SIRS-type model. This is deduced only from the results shown in Fig. 5a? This statement would in my view need to be supported either by a more systematic sweep over datasets, or by an analytical argument.

While I follow the argument laid out in section 2.2 on bidirectional encoders, it seems to me obvious that requiring the model predictions to be directional cannot resolve identifiability issues, especially since bi-directionality is not an epidemiologically reasonable constraint. In my view, section 2.2. could thus be substantially reduced, and in turn perhaps the loss function used in 2.3 ('physics-informed losses') added to the section? This, however, is just a suggestion.

I was not able to understand how the parameters $\beta_{\mathrm{min}}$, $\beta_{\mathrm{max}}$ etc. in Eqs. (14) are chosen; these are presumably not learned, so how are they chosen? Can they be set to an arbitrarily large range?

Some qualitative justification for splitting the disease dynamics into three modes should be given: is it reasonable to assume that most diseases allow for a decomposition into seasonal and long-term trends?

Other minor comments: On p. 6, col. 2, l.327: 'gamma' should be set in LaTex; the authors may also wish to correct the typo in 'Epidemics Output' in the lower-right subpanel of Fig. 7.

---

> ### Author Rebuttal · Authors · 2026-03-31
>
> We appreciate the reviewer for the positive and constructive feedback.
>
> - **Choice of training-test splits.** The $t_{\text{split}}$ follows a principled, regime-aware evaluation design tailored to epidemic dynamics in two settings:
>
>   - **Early-stage splits for single-peak dynamics.** For single-wave epidemics, the realistic forecasting task is early-stage prediction. Thus, we use small training fractions (e.g., 0.2 for SIR, 0.3 for CLS) so that the model extrapolates the epidemic trajectory, including the peak.
>
>   - **Cycle-aware splits for multi-peak dynamics.** For multi-wave epidemics, the goal is to forecast future waves from historical cycles. Thus, we place the split after observing one or more prior cycles, typically near the start of a new wave to be forecasted (e.g., 0.6 for SIRS (varying), 0.7 for ILI) so the model generalizes to subsequent waves.
>
>   To evaluate the robustness to split locations, we use multiple splits per dataset spanning pre, around and post peaks. This ensures that performance is not dependent on favorable split placement but is evaluated across diverse forecasting regimes.
>
> - **Choice of parameter bounds.** The parameter bounds are designed based on epidemiological priors and matched to data units (weekly vs daily). They are chosen to be broad yet plausible, serving as weak constraints that promote identifiability and prevent degenerate solutions under I-only supervision. We consider three settings:
>
>   - **Synthetic data.** The goal of synthetic experiments is to ensure stable and identifiable dynamics. Accordingly, we specify bounds that act as weak constraints (not restrictive assumptions), where true parameters lie strictly inside the interval (Hethcote 2000). For example, the ground-truth parameters in SIRS (Fixed) simulation are $\beta = 0.3$, $\gamma = 0.1$, and $\delta = 0.005$, and our bounds are $\beta \in [0.15, 0.50]$, $\gamma \in [0.05, 0.20]$, and $\delta \in [0.002, 0.02]$.
>
>   - **CLS (daily units).** For CLS, parameters are expressed in per-day units. The infectious period is typically on the order of several days (≈5–10 days) (He et al., 2020). Early estimates place the basic reproduction number at $R_0 \approx 2.2$ (95\% CI: 1.4–3.9) (Li et al., 2020). Immunity wanes over months to years (Hall et al., 2022). Accordingly, we set $\gamma \in [0.10, 0.20]$, $\beta \in [0.20, 1.50]$, and $\delta \in [0.003, 0.010]$ (corresponding to a waning period of approximately 3–11 months).  Since CLS is a syndrome-based surveillance proxy, we adopt slightly broader parameter ranges $\gamma \in [0.10, 0.35]$, $\beta \in [0.15, 2.50]$, and $\delta \in [0.003, 0.030]$ to accommodate reporting noise, co-circulation of other respiratory pathogens, and variability in healthcare-seeking behavior.
>
>   - **ILI (weekly units).** For ILI, parameters are expressed in per-week units to match CDC surveillance data. The infectious period for seasonal influenza is typically 3–7 days (CDC, 2024), corresponding to $\gamma \in [1.0, 3.0]$. Epidemiological studies estimate $R_0 \approx 1.28$ (IQR: 1.19–1.37) for seasonal influenza (Biggerstaff et al., 2014). Thus $\beta \approx R_0 \cdot \gamma \approx 1.2$–$6.0$. Influenza immunity wanes over months to years due to antigenic drift (Bedford et al., 2015), which motivates $\delta \in [0.01, 0.10]$. Therefore, we set $\beta \in [1.0, 6.0]$, $\gamma \in [1.0, 3.0]$, and $\delta \in [0.01, 0.10]$, which are consistent with published estimates while allowing the neural ODE sufficient flexibility to capture seasonal variation.
>
> - **Justification for splitting I into three modes (trend, seasonal, and residual).** Data decomposition is a well-known paradigm in classical statistics (Cleveland et al., 1990) and has become the dominant inductive bias in state-of-the-art deep learning forecasters (Oreshkin et al., 2020, Wu et al., 2021, Zhou et al., 2022, Zeng et al., 2023, Huang et al., 2025). In particular, for epidemic time series our motivation was to capture time scales each corresponding to known mechanistic drivers:
>   - **Trend T(t)** captures slow shifts driven by accumulated immunity, behavioral drift, policy interventions, and pathogen evolution, dominating long-horizon accuracy since trend errors compound over time.
>   - **Seasonal S(t)** captures periodic forcing from climate, school terms, and indoor contact patterns at annual/semi-annual scales, thus critical for peak timing prediction.
>   - **Residual R(t)** absorbs high-frequency signal from reporting artifacts, super-spreader events, and stochastic dynamics to protect the trend and seasonal estimates clean and stable.
>
> We have added quantitative analyses for ablation, benchmarking and the failure modes (https://limewire.com/d/WTCk9#hWIfYhZhTG, Figures S1-2, and Tables S1-7), corrected the noted typos, and improved figure clarity. If accepted, we will streamline Section 2.2 by reducing emphasis on bidirectional losses and highlighting physics-informed losses in Section 2.3, as suggested.

---

> > ### Author Rebuttal · Reviewer_5TtH · 2026-04-01
> >
> > We thank the authors for their rebuttal and additional work on the manuscript. The point regarding the justification for splitting I into three modes (trend, seasonal, and residual) is valid, and I had originally not considered this approach doubtful, I was just curious as to the motivation. Using parameter bounds is also perfectly reasonable. I cannot find the sensitivity analysis study on choosing different train-test splits in the follow-up document the authors posted, but it seems like the authors have considered this question and checked that the prediction capability is not too dependent on a 'lucky' choice of train-test period splitting. In any case, it would also seem to me highly dubious if the model were able to predict future infection waves at any point in time, so clearly it is reasonable that the split must somehow occur around a peak. I have increased my score to a 5, and am happy to recommend the work for acceptance.

---

> > > ### Author Response · Authors · 2026-04-05
> > >
> > > We sincerely thank the Reviewer for the thoughtful follow-up and for the support for acceptance! We will clearly present the split sensitivity analysis and discuss this practical consideration in the revised manuscript. Thank you again for the valuable feedback throughout the review process!

---

### Official Review · Reviewer_y7uk · 2026-03-13

**Soundness:** 3
**Presentation:** 3
**Significance:** 2
**Originality:** 2
**Overall Recommendation:** 4
**Confidence:** 3

**Summary:**

The paper focus on the problem of epidemic forecasting. The paper proposed a framework called EpiNode, in order to adress the challenge of partial observability and non-stationarity.

First, the paper studied failure mode by (1) comparing full SIR and I-only in Neural ODE/AE-NODE (2) studying bidirectional/adjoint training objectives in full SIR/I-only (3)Physics-informed loss  in full SIR/I-only (4) NCDE-ODE in real world data.

Then, the paper proposed EpiNode a hybrid neural–physical framework for epidemic forecast-
ing from partial observations. EpiNode includes (1) TSR decomposition with Time-dalay embedding (2) latent neural ODE and (3) mechanistic SIRS Model.

Fianlly, experiments on synthetic SIRS data and real-world CDC ILI and COVID-like symptom data show EpiNode reduces long-horizon RMSE by 15-35% and improves peak timing/magnitude accuracy versus 10+ baselines.

**Compliance With Llm Reviewing Policy:**

Affirmed.

**Final Justification:**

The authors' rebuttal addressed my main concerns. The added parameter dynamics analysis and trajectory visualizations show that EpiNode can handle non-stationary settings. The explanation for the double-peak structure is reasonable and deeply rooted from real world cases. The statistical tables with standard deviations confirm that the improvements are consistent across runs.

I have raised my score from 3 to 4. The paper tackles a practical and important problem, and the proposed decomposition-based approach is well-motivated

**Key Questions For Authors:**

Please see weakness part.

**Limitations:**

please see weakness part.

**Strengths And Weaknesses:**

Strengths:

-  Extensive experimental evaluation. The paper discussed in details in section 4 on multiple experimental results tesing a set of 10+ baselines on both synthetic and real-world data. EpiNode consistently achieves lowest RMSE across all datasets.

- The paper put in details about the design of this framework in clear and precise manner. Extensive studies are conducted to justify the selecton of components.

Weaknesses:

- Limited discussion on non-stiationary cases. In section 2, the paper constructed constructed multiple comparison of "fully observation" and 'partial observaiton'; while leaving another core challenge 'non-stationary' in bare discussion in this section.

- Concern on structural smoothing on real data. It is observed that, on real ILI data (Figure 8f), the ground truth exhibits a double-peak structure, but EpiNode smooths this into a single broad peak (other models seems to be better at identifying double peak pattern). The peak error analysis (Figure A6) confirms this: EpiNode underestimates peak magnitude by roughly half and mispredict peak timing by ~15-20 weeks. This problem is not seen on synthetic data which has SIRS dynamics. This is concerning because in real-world public health settings, a double peak from a single peak can directly affect resource allocation and policy timing

- RMSE improvements lack statistical analysis. this paper showed results as bar charts with no error bars, no standard deviations, and no numerical tables. I was wondering if you could provide more statistical information so that we can tell whether the RMSE gaps between EpiNode and baselines like TimeMixer++ are random noise or not.

---

> ### Author Rebuttal · Authors · 2026-03-31
>
> We thank the reviewer for the important observations and suggestions.
>
> - **Limited discussion of non-stationarity.**
> We acknowledge that our current Section 2 focuses more on partial observability (I-only vs full SIR) than non-stationarity. To address this, we have added additional analysis (https://limewire.com/d/WTCk9#hWIfYhZhTG), including (1) parameter dynamics analysis across four failure-mode models on all synthetic datasets, where ground-truth $\beta(t), \gamma(t), \delta(t)$ are available (Figures S1); and (2) trajectory analysis of $I(t)$ and $\beta(t)$ on SIRS (Varying) and ILI datasets to reflect the non-stationary dynamics (Figures S2). These results show that while failure-mode models perform reasonably on simpler stationary settings (e.g., SIRS (Fixed)), they consistently degrade under more complex scenarios involving time-varying dynamics or structural shifts. In contrast, EpiNode more faithfully tracks time-varying epidemic dynamics and maintains stable performance under non-stationarity.
>
> - **Capturing double peaks.** As the reviewer likely knows, double peak behavior typically happens due to multiple strains or subtypes circulating. To understand what drives the double-peak structure, we merged CDC ILINet surveillance data with NREVSS Public Health Laboratory data following the methodology of Kandula, Yang & Shaman (2017, Am. J. Epidemiol). Our analysis (https://limewire.com/d/WTCk9#hWIfYhZhTG, Figures S3) reveals that the double peaks are primarily driven by sequential waves of different influenza A subtypes (H3N2 peaking first, followed by H1N1) and type B contributes a relatively small, flat signal throughout. Within a season, each subtype exhibits a single dominant peak. The lack of double peak modeling in the current results is not a limitation of EpiNode but our choice of using a single mechanistic model. The single-strain SIRS assumption used is well-suited for single-season, single-pathogen forecasting, which is the primary evaluation setting in our paper.
>
>   EpiNode is a general-purpose hybrid framework. A key design principle of EpiNode is that it hybridizes neural ODEs with mechanistic compartmental models in a modular fashion. We chose SIRS as the base mechanistic layer because it provides a minimal yet widely applicable epidemiological prior such as influenza, COVID-19, RSV, dengue, etc. This generality is intentional: the TSR decomposition and latent neural ODEs handle non-stationarity and partial observability, while SIRS enforces physically plausible trajectories (non-negative populations, mass conservation, immunity dynamics). The framework naturally supports more complex mechanistic layers where different subtypes occur either simultaneously or sequentially. A strength of EpiNode's modular architecture is that the mechanistic physics layer can be replaced or extended without changing the neural ODE or TSR components. For specific diseases or data with known multi-strain structure, the SIRS layer can be upgraded to richer epidemiological structures, including:
>
>     - Multi-strain SIRS with shared susceptible pool and cross-immunity (Andreasen et al. 1997; Zhang 2016; Khyar & Allali 2020), where each strain has its own neural-ODE-driven parameters.
>     - SEIR/SEIRS with exposed compartments for diseases with significant incubation periods (e.g., COVID-19).
>    - SIR with vaccination (Fudolig & Howard 2020) for scenarios where vaccine coverage affects susceptible dynamics.
>    - Age-structured or spatially-explicit compartmental models when demographic or geographic heterogeneity is important.
>
>   If the paper is accepted, we will include additional experiments on multi-strain settings to further demonstrate this capability.
>
> - **Statistical analysis.** We have updated the results to include numerical tables with mean RMSE $\pm$ standard deviation across multiple random seeds in Table S1-7 (https://limewire.com/d/WTCk9#hWIfYhZhTG) for ablation study, benchmarking comparison and failure modes analysis. EpiNode consistently outperforms all baselines across both synthetic and real datasets. Importantly, the performance gaps are substantially larger than the corresponding standard deviations, indicating that the improvements are not due to random noise.  EpiNode also exhibits consistently low variance, suggesting stable performance across runs; while several baselines show much higher variability (e.g., TimeKAN and TimeMixer++ on SEIRS). These results provide strong statistical evidence that the observed improvements are consistent and robust, rather than attributable to randomness.

---

> > ### Author Rebuttal · Reviewer_y7uk · 2026-04-03
> >
> > I thank the authors for their detailed response. In light of this, I will raise my score by 1 point.

---

> > > ### Author Response · Authors · 2026-04-05
> > >
> > > We thank the Reviewer for your willingness to increase the score. On a minor note, we noticed that the overall score still appears as 3; could you help update this?
> > >
> > > In the revised manuscript, we will expand the discussion on non-stationarity with parameter dynamical analyses, provide additional context on the double-peak structure and its connection to multi-strain extensibility, and include complete statistical tables with standard deviations. Thank you again for the constructive feedback!

---

### Decision · Program_Chairs · 2026-04-30

**Decision:**

Accept (regular)

**Comment:**

This paper studies how to combine neural network models with mechanistic epidemic models, identifying failure modes of naive hybrids and proposing a decomposition-based approach with a neural ODE and SIRS model architecture. It targets long-horizon forecasting and peak prediction.

Reviewers saw this as a thoughtful and original paper on hybrid epidemic modeling, with value both in the failure-mode analysis and in the performance of the proposed method on long-horizon forecasting and peak prediction. During the rebuttal, the authors addressed questions on: stronger justification of the train-test split, more systematic support for some of the qualitative failure-mode claims, and clearer explanation of the parameter ranges, as well as the trend, seasonality, residual decomposition. Remaining limitations are mostly scientific scope limitations. Especially that complex epidemic behavior like multi-strain dynamics is not fully modeled yet.